# Amortizing intractable inference in diffusion models for vision, language, and control

**Siddarth Venkatraman**[*]
Mila, Université de Montréal

**Moksh Jain**[*]
Mila, Université de Montréal

**Luca Scimeca**[*]
Mila, Université de Montréal

**Minsu Kim**[*]
Mila, Université de Montréal
KAIST

**Marcin Sendera**[*]
Mila, Université de Montréal
Jagiellonian University

**Mohsin Hasan**
Mila, Université de Montréal

**Luke Rowe**
Mila, Université de Montréal

**Sarthak Mittal**
Mila, Université de Montréal

**Pablo Lemos**
Mila, Université de Montréal
Ciela Institute
Dreamfold

**Emmanuel Bengio**
Recursion

**Alexandre Adam**
Mila, Université de Montréal
Ciela Institute

**Jarrid Rector-Brooks**
Mila, Université de Montréal
Dreamfold

**Yoshua Bengio**
Mila, Université de Montréal
CIFAR

**Glen Berseth**
Mila, Université de Montréal
CIFAR

**Nikolay Malkin**
Mila, Université de Montréal
University of Edinburgh

$\left\{\begin{array}{l}\texttt{siddarth.venkatraman,moksh.jain,luca.scimeca}\\\texttt{minsu.kim,marcin.sendera,...,nikolay.malkin}\end{array}\right\}$`@mila.quebec`

## Abstract

Diffusion models have emerged as effective distribution estimators in vision, language, and reinforcement learning, but their use as priors in downstream tasks poses an intractable posterior inference problem. This paper studies *amortized* sampling of the posterior over data, $\mathbf{x} \sim p^{\text{post}}(\mathbf{x}) \propto p(\mathbf{x})r(\mathbf{x})$, in a model that consists of a diffusion generative model prior $p(\mathbf{x})$ and a black-box constraint or likelihood function $r(\mathbf{x})$. We state and prove the asymptotic correctness of a data-free learning objective, *relative trajectory balance*, for training a diffusion model that samples from this posterior, a problem that existing methods solve only approximately or in restricted cases. Relative trajectory balance arises from the generative flow network perspective on diffusion models, which allows the use of deep reinforcement learning techniques to improve mode coverage. We illustrate the broad potential of unbiased inference of arbitrary posteriors under diffusion priors across a collection of experiments: in vision (classifier guidance), language (infilling under a discrete diffusion LLM), and multimodal data (text-to-image generation). Beyond generative modeling, we apply relative trajectory balance to the problem of continuous control with a score-based behavior prior, achieving state-of-the-art results on benchmarks in offline reinforcement learning. Code is available at [this link](#).

38th Conference on Neural Information Processing Systems (NeurIPS 2024).

Table 1: Sources of diffusion priors and constraints.

| Domain | Prior $p(\mathbf{x})$ | Constraint $r(\mathbf{x})$ | Posterior |
|---|---|---|---|
| Conditional image generation (§3.1) | Image diffusion model $p(\mathbf{x})$ | Classifier likelihood $p(c \mid \mathbf{x})$ | Class-conditional distribution $p(\mathbf{x} \mid c)$ |
| Text-to-image generation (§3.2) | Text-to-image foundation model | RLHF reward model | Aligned text-to-image model |
| Language infilling (§3.3) | Discrete diffusion model | Autoregressive completion likelihood | Infilling distribution |
| Offline RL policy extraction (§3.4) | Diffusion model as behavior policy | Boltzmann dist. of $Q$-function | Optimal KL-constrained policy |

# 1 Introduction

Diffusion models [68, 27, 72] are a powerful class of hierarchical generative models, used to model complex distributions over images [51, 12, 63], text [4, 13, 40, 24, 23, 43], and actions in reinforcement learning [30, 83, 32] to name a few. In each of these domains, downstream problems require sampling product distributions, where a pretrained diffusion model serves as a prior $p(\mathbf{x})$ that is multiplied by an auxiliary constraint $r(\mathbf{x})$. For example, if $p(\mathbf{x})$ is a prior over images defined by a diffusion model, and $r(\mathbf{x}) = p(c \mid \mathbf{x})$ is the likelihood that an image $\mathbf{x}$ belongs to class $c$, then class-conditional image generation requires sampling from the Bayesian posterior $p(\mathbf{x} \mid c) \propto p(\mathbf{x})p(c \mid \mathbf{x})$. In offline reinforcement learning, if $\mu(a \mid s)$ is a conditional diffusion model over actions serving as a behavior policy, KL-constrained policy improvement [55, 44] requires sampling from the normalized product of $\mu(a \mid s)$ with a Boltzmann distribution defined by a $Q$-function, $\pi^*(a \mid s) \propto \mu(a \mid s) \exp(\beta Q(s, a))$. In language modeling, various conditional generation problems [43, 23, 29] amount to posterior sampling under a discrete diffusion model prior. Table 1 summarizes four such problems that the proposed method improves upon prior work.

The hierarchical nature of the generative process in diffusion models, which generate samples from $p(\mathbf{x})$ by a deep chain of stochastic transformations, makes exact sampling from posteriors $p(\mathbf{x})r(\mathbf{x})$ under a black-box function $r(\mathbf{x})$ intractable. Common solutions to this problem involve inference techniques based on linear approximations [73, 33, 31, 11] or stochastic optimization [22, 48]. Others estimate the 'guidance' term – the difference in drift functions between the diffusion models sampling the prior and posterior – by training a classifier on noised data [12], but when such data is not available, one must resort to approximations or Monte Carlo estimates [70, 14, 10], which are challenging to scale to high-dimensional problems. Reinforcement learning methods that have recently been proposed for this problem [8, 16] are biased and prone to mode collapse (Fig. 1).

**Contributions.** Inspired by recent techniques in training diffusion models to sample distributions defined by unnormalized densities [89, 62, 78, 65], we propose an asymptotically unbiased training objective, called relative trajectory balance (RTB), for training diffusion models that sample from posterior distributions under a diffusion model prior (§2.2). RTB is derived from the perspective of diffusion models as continuous generative flow networks [38]. This perspective also allows us to freely leverage off-policy training, when data with high density under the posterior is available (§2.3). RTB can be applied to iterative generative processes beyond standard diffusion models: our methods generalize to discrete diffusion models and extend existing methods for autoregressive language models (§2.4).

Our experiments demonstrate the versatility of our approach in a variety of domains:

- In **vision**, we show that RTB achieves competitive classifier-guided image generation for unconditional diffusion vision priors (§3.1) and can be used to improve caption-conditioned generation under text-to-image foundation model priors (§3.2).
- In **language modeling**, we report strong results for infilling tasks with discrete diffusion language models (§3.3).
- Finally, we show that RTB achieves state-of-the-art results on **continuous control** benchmarks that leverage score-based behavior priors (§3.4).

# 2 Learning posterior samplers with diffusion priors

We consider the problem of posterior inference under a prior given by a hierarchical generative model. In this section, we present the mathematical setting (§2.1), our proposed RTB objective (§2.2), and training methods for RTB (§2.3). We will first discuss the case of a diffusion prior over $\mathbb{R}^d$, and later discuss how the methods generalize to arbitrary hierarchical priors (§2.4).

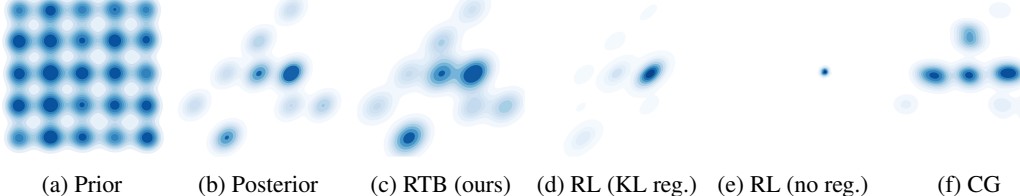

| (a) Prior | (b) Posterior | (c) RTB (ours) | (d) RL (KL reg.) | (e) RL (no reg.) | (f) CG |

Figure 1: Sampling densities learned by various posterior inference methods. The prior is a diffusion model sampling a mixture of 25 Gaussians (a) and the posterior is the product of the prior with a constraint that masks all but 9 of the modes (b). Our method (RTB) samples close to the true posterior (c). RL methods with tuned KL regularization yield inaccurate inference (d), while without KL regularization, they mode-collapse (e). A classifier guidance (CG) approximation (f) results in biased outcomes. For details, see §C.

## 2.1 Background and setting: Diffusion models as hierarchical generative models

A denoising diffusion model generates data $\mathbf{x}_1$ by a Markovian generative process:

$$(noise) \quad \mathbf{x}_0 \rightarrow \mathbf{x}_{\Delta t} \rightarrow \mathbf{x}_{2\Delta t} \rightarrow \ldots \rightarrow \mathbf{x}_1 = \mathbf{x} \quad (data), \tag{1}$$

where $\Delta t = \frac{1}{T}$ and $T$ is the number of discretization steps.[1] The initial distribution $p(\mathbf{x}_0)$ is fixed (typically to $\mathcal{N}(\mathbf{0}, \boldsymbol{I})$) and the transition from $\mathbf{x}_{t-1}$ to $\mathbf{x}_t$ is modeled as a Gaussian perturbation with time-dependent variance:

$$p(\mathbf{x}_{t+\Delta t} \mid \mathbf{x}_t) = \mathcal{N}(\mathbf{x}_{t+\Delta t} \mid \mathbf{x}_t + u_t(\mathbf{x}_t)\Delta t, \sigma_t^2 \Delta t \boldsymbol{I}). \tag{2}$$

The scaling of the mean and variance by $\Delta t$ is insubstantial for fixed $T$, but ensures that the diffusion process is well-defined in the limit $T \rightarrow \infty$ assuming regularity conditions on $u_t$ [53, 64]. The process given by (1, 2) is then identical to Euler-Maruyama integration of the stochastic differential equation (SDE) $d\mathbf{x}_t = u_t(\mathbf{x}_t)\, dt + \sigma_t\, d\mathbf{w}_t$.

The likelihood of a denoising trajectory $\mathbf{x}_0 \rightarrow \mathbf{x}_{\Delta t} \rightarrow \cdots \rightarrow \mathbf{x}_1$ factors as

$$p(\mathbf{x}_0, \mathbf{x}_{\Delta t}, \ldots, \mathbf{x}_1) = p(\mathbf{x}_0) \prod_{i=1}^{T} p(\mathbf{x}_{i\Delta t} \mid \mathbf{x}_{(i-1)\Delta t}) \tag{3}$$

and defines a marginal density over the data space:

$$p(\mathbf{x}_1) = \int p(\mathbf{x}_0, \mathbf{x}_{\Delta t}, \ldots, \mathbf{x}_1)\, d\mathbf{x}_0\, d\mathbf{x}_{\Delta t} \ldots d\mathbf{x}_{1-\Delta t}. \tag{4}$$

A reverse-time process, $\mathbf{x}_1 \rightarrow \mathbf{x}_{1-\Delta t} \rightarrow \cdots \rightarrow \mathbf{x}_0$, with densities $q$, can be defined analogously, and similarly defines a conditional density over trajectories:

$$q(\mathbf{x}_0, \mathbf{x}_{\Delta t}, \ldots, \mathbf{x}_{1-\Delta t} \mid \mathbf{x}_1) = \prod_{i=1}^{T} q(\mathbf{x}_{(i-1)\Delta t} \mid \mathbf{x}_{i\Delta t}). \tag{5}$$

In the training of diffusion models, as discussed below, the process $q$ is typically fixed to a simple distribution (usually a discretized Ornstein-Uhlenbeck process), and the result of training is that $p$ and $q$ are close as distributions over trajectories.

**Diffusion model training as divergence minimization.** Diffusion models parametrize the drift $u_t(\mathbf{x}_t)$ in (Equation 2) as a neural network $u(\mathbf{x}_t, t; \theta)$ with parameters $\theta$ and taking $\mathbf{x}_t$ and $t$ as input. We denote the distributions over trajectories induced by (Equation 3, Equation 4) by $p_\theta$ to show their dependence on the parameter.

In the most common setting, diffusion models are trained to maximize the likelihood of a dataset. In the notation above, this corresponds to assuming $q(\mathbf{x}_1)$ is fixed to an empirical measure (with the

---

[1]The time indexing suggestive of an SDE discretization is used for consistency with the diffusion samplers literature [89, 65]. The indexing $\mathbf{x}_T \rightarrow \mathbf{x}_{T-1} \rightarrow \cdots \rightarrow \mathbf{x}_0$ is often used for diffusion models trained from data.

points of a training dataset $\mathcal{D}$ assumed to be i.i.d. samples from $q(\mathbf{x}_1)$). Training minimizes with respect to $\theta$ the divergence between the processes $q$ and $p_\theta$:

$$D_{\mathrm{KL}}(q(\mathbf{x}_0, \mathbf{x}_{\Delta t}, \ldots, \mathbf{x}_1) \| p_\theta(\mathbf{x}_0, \mathbf{x}_{\Delta t}, \ldots, \mathbf{x}_1)) \tag{6}$$
$$= D_{\mathrm{KL}}(q(\mathbf{x}_1) \| p_\theta(\mathbf{x}_1)) + \mathbb{E}_{\mathbf{x}_1 \sim q(\mathbf{x}_1)} D_{\mathrm{KL}}(q(\mathbf{x}_0, \mathbf{x}_{\Delta t}, \ldots, \mathbf{x}_{1-\Delta t} \mid \mathbf{x}_1) \| p_\theta(\mathbf{x}_0, \mathbf{x}_{\Delta t}, \ldots, \mathbf{x}_{1-\Delta t} \mid \mathbf{x}_1))$$
$$\geq D_{\mathrm{KL}}(q(\mathbf{x}_1) \| p_\theta(\mathbf{x}_1)) = \mathbb{E}_{\mathbf{x}_1 \sim q(\mathbf{x}_1)}[-\log p_\theta(\mathbf{x}_1)] + \mathrm{const.}$$

where the inequality – an instance of the data processing inequality for the KL divergence – shows that minimizing the divergence between distributions over trajectories is equivalent to maximizing a lower bound on the data log-likelihood under the model $p_\theta$.

As shown in [71], minimization of the KL in (Equation 6) is essentially equivalent to the traditional approach to training diffusion models via denoising score matching [80, 68, 27]. Such training exploits that for typical choices of the noising process $q$, the optimal $u_t(\mathbf{x}_t)$ can be expressed in terms of the Stein score of $q(\mathbf{x}_1)$ convolved with a Gaussian, allowing an efficient stochastic regression objective for $u_t$. For full generality of our exposition for arbitrary iterative generative processes, we prefer to think of (Equation 6) as the primal objective and denoising score matching as an efficient means of minimizing it.

**Trajectory balance and distribution-matching training.** From (Equation 6) we also see that the bound is tight if the conditionals of $p_\theta$ and $q$ on $\mathbf{x}_1$ coincide, *i.e.*, $q$ is equal to the posterior distribution of $p$ conditioned on $\mathbf{x}_1$. Indeed, the model $p_\theta$ minimizes (Equation 6) for a distribution with continuous density $q(\mathbf{x}_1)$ if and only if, for all denoising trajectories,

$$p_\theta(\mathbf{x}_0, \mathbf{x}_{\Delta t}, \ldots, \mathbf{x}_1) = q(\mathbf{x}_1)q(\mathbf{x}_0, \mathbf{x}_{\Delta t}, \ldots, \mathbf{x}_{1-\Delta t} \mid \mathbf{x}_1). \tag{7}$$

This was named the *trajectory balance (TB) constraint* by [38] – by analogy with a constraint for discrete-space iterative sampling [46] – and is a time-discretized version of a constraint used for enforcing equality of continuous-*time* path space measures in [52] (see [7] for asymptotic analysis).

In [61, 38], the constraint (7) was used for the training of diffusion models in a *data-free* setting, where instead of i.i.d. samples from $q(\mathbf{x}_1)$ one has access to a (possibly unnormalized) density $q(\mathbf{x}_1) = e^{-\mathcal{E}(\mathbf{x}_1)}/Z$ from which one wishes to sample. These objectives minimize the squared log-ratio between the two sides of (7), which allows the trajectories $\mathbf{x}_0 \to \mathbf{x}_{\Delta t} \to \cdots \to \mathbf{x}_1$ used for training to be sampled from any training distribution, such as 'exploratory' modifications of $p_\theta$ or trajectories found by local search (MCMC) in the target space. The flexibility of off-policy exploration that this allows was studied by [65]. Such objectives contrast with on-policy, simulation-based approaches that require differentiating through the sampling process [*e.g.*, 89, 78, 6, 79].

## 2.2 Intractable inference under diffusion priors

Consider a diffusion model $p_\theta$, defining a marginal density $p_\theta(\mathbf{x}_1)$, and a positive constraint function $r : \mathbb{R}^d \to \mathbb{R}_{>0}$. We are interested in training a diffusion model $p_\phi^{\mathrm{post}}$, with drift function $u_\phi^{\mathrm{post}}$, that would sample the product distribution $p^{\mathrm{post}}(\mathbf{x}_1) \propto p_\theta(\mathbf{x}_1)r(\mathbf{x}_1)$. If $r(\mathbf{x}_1) = p(\mathbf{y} \mid \mathbf{x}_1)$ is a conditional distribution over another variable $\mathbf{y}$, then $p^{\mathrm{post}}$ is the Bayesian posterior $p_\theta(\mathbf{x}_1 \mid \mathbf{y})$.

Because samples from $p^{\mathrm{post}}(\mathbf{x}_1)$ are not assumed to be available, one cannot directly train $p$ using the objective (6). Nor can one directly apply objectives for distribution-matching training, such as those that enforce (7), since the marginal $p_\theta(\mathbf{x}_1)$ is not available. However, we make the following observation (proof in §A).

**Proposition 1** (Relative TB constraint)**.** *If $p_\theta$, $p_\phi^{\mathrm{post}}$, and the scalar $Z_\phi$ jointly satisfy the* relative trajectory balance (RTB) constraint

$$Z_\phi \cdot p_\phi^{\mathrm{post}}(\mathbf{x}_0, \mathbf{x}_{\Delta t}, \ldots, \mathbf{x}_1) = r(\mathbf{x}_1)p_\theta(\mathbf{x}_0, \mathbf{x}_{\Delta t}, \ldots, \mathbf{x}_1) \tag{8}$$

*for every denoising trajectory $\mathbf{x}_0 \to \mathbf{x}_{\Delta t} \to \cdots \to \mathbf{x}_1$, then $p_\phi^{\mathrm{post}}(\mathbf{x}_1) \propto p_\theta(\mathbf{x}_1)r(\mathbf{x}_1)$, i.e., the diffusion model $p_\phi^{\mathrm{post}}$ samples the posterior distribution. Furthermore, if $p_\theta$ also satisfies the TB constraint (7) with respect to the noising process $q$ and some target density $q(\mathbf{x}_1)$, then $p_\phi^{\mathrm{post}}$ satisfies the TB constraint with respect to the target density $q^{\mathrm{post}}(\mathbf{x}_1) \propto q(\mathbf{x}_1)r(\mathbf{x}_1)$, and $Z = \int q(\mathbf{x}_1)r(\mathbf{x}_1)\,d\mathbf{x}_1$.*

Note that the two joints appearing in (8) are defined as products over transitions, via (3).

**Relative trajectory balance as a loss.** Analogously to the conversion of the TB constraint (7) into a trajectory-dependent training objective in [46, 38], we define the *relative trajectory balance loss* as the discrepancy between the two sides of (8), seen as a function of the vector $\phi$ that parametrizes the posterior diffusion model and the scalar $Z_\phi$ (parametrized via $\log Z_\phi$ for numerical stability):

$$\mathcal{L}_{\text{RTB}}(\mathbf{x}_0 \to \mathbf{x}_{\Delta t} \to \cdots \to \mathbf{x}_1; \phi) := \left( \log \frac{Z_\phi \cdot p_\phi^{\text{post}}(\mathbf{x}_0, \mathbf{x}_{\Delta t}, \ldots, \mathbf{x}_1)}{r(\mathbf{x}_1) p_\theta(\mathbf{x}_0, \mathbf{x}_{\Delta t}, \ldots, \mathbf{x}_1)} \right)^2. \tag{9}$$

Optimizing this objective to 0 for all trajectories ensures that (8) is satisfied. While the RTB constraint (8) has a similar form to TB (7), RTB involves the ratio of two denoising processes, while TB involves the ratio of a forward and a backward process. However, the name 'relative TB' is justified by interpreting the densities in a TB constraint relative to a measure defined by the prior model; see §2.4.

If we assume $p_\theta(\mathbf{x}_0) = p_\phi^{\text{post}}(\mathbf{x}_0)$ are fixed (*e.g.*, to a standard normal), then (9) reduces to

$$\left( \log \frac{Z_\phi}{r(\mathbf{x}_1)} + \sum_{i=1}^{T} \log \frac{p_\phi^{\text{post}}(\mathbf{x}_{i\Delta t} \mid \mathbf{x}_{(i-1)\Delta t})}{p_\theta(\mathbf{x}_{i\Delta t} \mid \mathbf{x}_{(i-1)\Delta t})} \right)^2. \tag{10}$$

Notably, the gradient of this objective with respect to $\phi$ does not require differentiation (backpropagation) into the sampling process that produced a trajectory $\mathbf{x}_0 \to \cdots \to \mathbf{x}_1$. This offers two advantages over on-policy simulation-based methods: (1) the ability to optimize $\mathcal{L}_{\text{RTB}}$ as an off-policy objective, *i.e.*, sampling trajectories for training from a distribution different from $p_\phi^{\text{post}}$ itself, as discussed further in §2.3; (2) backpropagating only to a subset of the summands in (10), when computing and storing gradients for all steps in the trajectory is prohibitive for large diffusion models (see §H.1).

**Comparison with classifier guidance.** It is interesting to contrast the RTB training objective with the technique of *classifier guidance* [12] used for some problems of the same form. If $r(\mathbf{x}_1) = p(\mathbf{y} \mid \mathbf{x}_1)$ is a conditional likelihood, classifier guidance relies upon writing $u_t(\mathbf{x}_t) - u_t^{\text{post}}(\mathbf{x}_t)$ explicitly in terms of $\nabla_{\mathbf{x}_t} \log p(\mathbf{y} \mid \mathbf{x}_t)$, by combining the expression of the optimal drift $u_t$ in terms of the score of the target distribution convolved with a Gaussian (cf. §2.1), with the 'Bayes' rule' for the Stein score: $\nabla_{\mathbf{x}_t} \log p(\mathbf{x}_t \mid \mathbf{y}) = \nabla_{\mathbf{x}_t} \log p(\mathbf{x}_t) + \nabla_{\mathbf{x}_t} \log p(\mathbf{y} \mid \mathbf{x}_t)$.

Classifier guidance gives the *exact* solution for the posterior drift when a differentiable classifier on noisy data, $p(\mathbf{y} \mid \mathbf{x}_t) = \int p(\mathbf{y} \mid \mathbf{x}_1) p(\mathbf{x}_1 \mid \mathbf{x}_t) \, d\mathbf{x}_1$, is available. Unfortunately, such a classifier is not, in general, tractable to derive from the classifier on noiseless data, $p(\mathbf{y} \mid \mathbf{x}_1)$, and cannot be learned without access to unbiased data samples. RTB is an asymptotically unbiased objective that recovers the difference in drifts (and thus the gradient of the log-convolved likelihood) in a data-free manner.

## 2.3 Training, parametrization, and conditioning

**Training and exploration.** The choice of which trajectories we use to take gradient steps with the RTB loss can have a large impact on sample efficiency. In *on-policy* training, we use the current policy $p_\phi^{\text{post}}$ to generate trajectories $\tau = (\mathbf{x}_0 \to \ldots \to \mathbf{x}_1)$, evaluate the reward $\log r(\mathbf{x}_1)$ and the likelihood of $\tau$ under $p_\theta$, and a gradient updates on $\phi$ to minimize $\mathcal{L}_{\text{RTB}}(\tau; \phi)$.

However, on-policy training may be insufficient to discover the modes of the posterior distribution. In this case, we can perform *off-policy* exploration to ensure mode coverage. For instance, given samples $\mathbf{x}_1$ that have high density under the target distribution, we can sample *noising* trajectories $\mathbf{x}_1 \leftarrow \mathbf{x}_{1-\Delta t} \leftarrow \ldots \leftarrow \mathbf{x}_0$ starting from these samples and use such trajectories for training. Another effective off-policy training technique uses replay buffers. We expect the flexibility of mixing on-policy training with off-policy exploration to be a strength of RTB over on-policy RL methods, as was shown for distribution-matching training of diffusion models in [65].

**Conditional constraints and amortization.** Above we derived and proved the correctness of the RTB objective for an arbitrary positive constraint $r(\mathbf{x}_1)$. If the constraints depend on other variables $\mathbf{y}$ – for example, $r(\mathbf{x}_1; \mathbf{y}) = p(\mathbf{y} \mid \mathbf{x}_1)$ – then the posterior drift $u_\phi^{\text{post}}$ can be conditioned on $\mathbf{y}$ and the learned scalar $\log Z_\phi$ replaced by a model taking $\mathbf{y}$ as input. Such conditioning achieves amortized inference and allows generalization to new $\mathbf{y}$ not seen in training. Similarly, all of the preceding discussion easily generalizes to *priors* that are conditioned on some context variable.

**Efficient parametrization and Langevin inductive bias.** Because the deep features learned by the prior model $u_\theta$ are expected to be useful in expressing the posterior drift $u_\phi^{\text{post}}$, we can choose to initialize $u_\phi^{\text{post}}$ as a copy of $u_\theta$ and to fine-tune it, possibly in a parameter-efficient way (as described in each section of §3). This choice is inspired by the method of amortizing inference in large language models by fine-tuning a prior model to sample an intractable posterior [29].

Furthermore, if the constraint $r(\mathbf{x}_1)$ is differentiable, we can impose an inductive bias on the posterior drift similar to the one introduced for diffusion samplers of unnormalized target densities in [89] and shown to be useful for off-policy methods in [65]. namely, we write

$$u_\phi^{\text{post}}(\mathbf{x}_t, t) = \text{NN}_1(\mathbf{x}_t, t; \phi) + \text{NN}_2(\mathbf{x}_t, t, \phi)\nabla_{\mathbf{x}_t} \log r(\mathbf{x}_t), \tag{11}$$

where $\text{NN}_1$ and $\text{NN}_2$ are neural networks outputting a vector and a scalar, respectively. This parametrization allows the constraint to provide a signal to guide the sampler at intermediate steps.

**Stabilizing the loss.** We propose two simple design choices for stabilizing RTB training. First, the loss in (9) can be replaced by the empirical *variance* over a minibatch of the quantity inside the square, which removes dependence on $\log Z_\phi$ and is especially useful in conditional settings, consistent with the findings of [65]. This amounts to a relative variant of the VarGrad objective [61] (see (23) in §G). Second, we employ loss clipping: to reduce sensitivity to an imperfectly fit prior model, we do not perform updates on trajectories where the loss is close to 0 (see §E,§F).

## 2.4 Generative flow networks and extension to other hierarchical processes

**RTB as TB under the prior measure.** The theoretical foundations for continuous generative flow networks [38] establish the correctness of enforcing constraints such as trajectory balance (7) for training sequential samplers, such as diffusion models, to match unnormalized target densities. While we have considered Gaussian transitions and identified transition kernels with their densities with respect to the Lebesgue measure over $\mathbb{R}^d$, these foundations generalize to more general *reference measures*. In §B, we show how the RTB constraint can be recovered as a special case of the TB constraint for a certain choice of reference measure derived from the prior.

**Extension to arbitrary sequential generation.** While our discussion was focused on diffusion models for continuous spaces, the RTB objective can be applied to any Markovian sequential generative process, in particular, one that can be formulated as a generative flow network in the sense of [5, 38]. This includes, in particular, generative models that generate objects by a sequence of discrete steps, including autoregressive models and discrete diffusion models. In the case of discrete diffusion, where the intermediate latent variables $\mathbf{x}_t$ lie not in $\mathbb{R}^d$ but in the space of sequences, one simply replaces the Gaussian transition densities by transition probability *masses* in the RTB constraint (8) and objective (9). In the case of autoregressive models, where only one sequence of steps can generate any given object, the backward process $q$ becomes trivial, and the RTB constraint for a model $p_\phi^{\text{post}}$ to sample a sequence $\mathbf{x}$ from a distribution with density $r(\mathbf{x})p_\theta(\mathbf{x})$ is simply $Z_\phi p_\phi^{\text{post}}(\mathbf{x}) = r(\mathbf{x})p_\theta(\mathbf{x})$ for all sequences $\mathbf{x}$. We note that a sub-trajectory generalization of this objective was used in [29] to amortize intractable inference in autoregressive language models.

# 3 Experiments

In this section, we present empirical results to validate the efficacy of relative trajectory balance. Our experiments are designed to demonstrate the wide applicability of RTB to sample from posteriors for diffusion priors with arbitrary rewards on vision, language, and continuous control tasks.

## 3.1 Class-conditional posterior sampling from unconditional diffusion priors

We evaluate RTB in a classifier-guided visual task where we wish to learn a diffusion posterior $p_\phi^{\text{post}}(\mathbf{x} \mid c) \propto p_\theta(\mathbf{x})p(c \mid \mathbf{x})$ given a pretrained diffusion prior $p_\theta(\mathbf{x})$ and a classifier $r(\mathbf{x}) = p(c \mid \mathbf{x})$.

**Setup.** We consider two 10-class image datasets, MNIST and CIFAR-10, using off-the-shelf unconditional diffusion priors from [27] and standard classifiers $p(c \mid \mathbf{x})$ for both datasets. We perform parameter-efficient fine-tuning of $p_\phi^{\text{post}}$, initialized as a copy of the prior $p_\theta$, using the RTB objective (see §E.1 for details). The RTB objective is optimized on trajectories sampled

Table 2: Classifier-guided posterior sampling with pretrained unconditional diffusion priors. We report the mean±std of each metric computed across all relevant classes for each experiment set, and highlight ±5% from highest/lower experimental value. The FID is computed between learned posterior samples and the true samples from the class in question. DP and LGD-MC fail to appropriately model the posterior distribution (high average $\log r(\mathbf{x})$) while DDPO mode-collapses. RTB achieves comparable or superior performance to all other baselines, optimally balancing high reward and diversity as measured by FID. See Table E.1 for conditional variants.

| Dataset → | MNIST | | | MNIST even/odd | | | CIFAR-10 | | |
|---|---|---|---|---|---|---|---|---|---|
| Algorithm ↓ Metric → | $\mathbb{E}[\log r(\mathbf{x})]$ (↑) | FID (↓) | Diversity (↑) | $\mathbb{E}[\log r(\mathbf{x})]$ (↑) | FID (↓) | Diversity (↑) | $\mathbb{E}[\log r(\mathbf{x})]$ (↑) | FID (↓) | Diversity (↑) |
| DPS | $-2.1597_{\pm 0.423}$ | $1.2913_{\pm 0.410}$ | $0.1609_{\pm 0.000}$ | $-1.2270_{\pm 0.202}$ | $1.1498_{\pm 0.182}$ | $0.1713_{\pm 0.000}$ | $-3.6025_{\pm 0.503}$ | $0.7371_{\pm 0.216}$ | $0.2738_{\pm 0.000}$ |
| LGD−MC | $-2.1389_{\pm 0.480}$ | $1.2873_{\pm 0.412}$ | $0.1600_{\pm 0.000}$ | $-1.1720_{\pm 0.199}$ | $1.1445_{\pm 0.184}$ | $0.1600_{\pm 0.000}$ | $-3.0988_{\pm 0.359}$ | $0.7402_{\pm 0.214}$ | $0.2743_{\pm 0.000}$ |
| DDPO | $-1.5_{\pm 4.7} \times 10^{-3}$ | $1.5822_{\pm 0.583}$ | $0.1350_{\pm 0.005}$ | $-8.6_{\pm 12.3} \times 10^{-11}$ | $1.8024_{\pm 0.423}$ | $0.1314_{\pm 0.002}$ | $-2.7_{\pm 8.5} \times 10^{-4}$ | $1.7686_{\pm 0.589}$ | $0.1575_{\pm 0.015}$ |
| DPOK | $-0.1379_{\pm 0.225}$ | $1.2063_{\pm 0.316}$ | $0.1442_{\pm 0.004}$ | $-0.0783_{\pm 0.082}$ | $1.2536_{\pm 0.206}$ | $0.1631_{\pm 0.007}$ | $-2.4414_{\pm 3.266}$ | $0.5316_{\pm 0.157}$ | $0.2415_{\pm 0.024}$ |
| **RTB (ours)** | $-0.1734_{\pm 0.194}$ | $1.1823_{\pm 0.288}$ | $0.1474_{\pm 0.003}$ | $-0.1816_{\pm 0.175}$ | $1.1794_{\pm 0.171}$ | $0.1679_{\pm 0.004}$ | $-2.1625_{\pm 0.879}$ | $0.4717_{\pm 0.138}$ | $0.2440_{\pm 0.011}$ |

Prior    Posterior (seven)    Posterior (even)    Prior    Posterior (dog)

MNIST                                          CIFAR-10

Figure 2: Samples from RTB fine-tuned diffusion posteriors.

on-policy from the current posterior model. We compare RTB with two RL-based fine-tuning techniques derived from DPOK [16] and DDPO [8] and with two classifier guidance baselines, namely DPS [11], and LGD-MC [70]. We consider three experimental settings: MNIST single-digit posterior (learning to sample images of each digit class $c$), CIFAR-10 single-class posterior (analogous to the previous), and MNIST multi-digit posterior. The latter is a multimodal posterior, for which we set $r(\mathbf{x}) = \max_{i \in \{0,2,4,6,8\}} p(c = i \mid \mathbf{x})$ to generate even digits, and similarly for odd digits.

**Results.** Samples from the RTB-fine-tuned posterior models are shown in Fig. 2. In Table 2 we report mean±std of various metrics across all trained posteriors. We observe that models fine-tuned with RTB generate class samples with both the highest diversity (highest mean pairwise cosine distance in Inceptionv3 feature space) and closeness to true samples of the target classes (FID), while achieving high expected $\log r(\mathbf{x})$. Pure RL fine-tuning (no KL regularization) displays mode collapse characteristics, achieving high rewards in exchange for significantly poorer diversity and FID scores (see also Fig. E.1). Classifier-guidance-based methods, like DP and LGD-MC, exhibit high diversity, but fail to appropriately model the posterior distribution (lowest $\log r(\mathbf{x})$). Additional results can be found in §E.2.

## 3.2 Fine-tuning a text-to-image diffusion model

Diffusion models for text-conditional image generation [*e.g.* 63] can struggle to consistently generate images $\mathbf{x}$ that adhere to complex prompts $\mathbf{z}$, for example, those that involve composing multiple objects (*e.g.*, "A cat and a dog") or specify "unnatural" appearances (*e.g.*, "A green-colored rabbit"). Fine-tuning pretrained text-to-image diffusion models $p_\theta(\mathbf{x}_1 \mid \mathbf{z})$ as RL policies to maximize some reward $r(\mathbf{x}_1, \mathbf{z})$ based on human preferences has become the standard approach to tackle this issue [8, 16, 77]. Simply maximizing the reward function can result in mode collapse as well as over-optimization of the reward. This is typically handled by constraining the fine-tuned model $\tilde{p}$ to be close to the prior $p$:

$$\underset{\tilde{p}}{\arg\max} \, \mathbb{E}_{\tilde{p}(\mathbf{x}_1|\mathbf{z})}[r(\mathbf{x}_1, \mathbf{z})], \quad D_{\mathrm{KL}}[\tilde{p}(\mathbf{x}_1 \mid \mathbf{z}) \, \| \, p(\mathbf{x}_1 \mid \mathbf{z})] \le \epsilon. \tag{12}$$

The optimal $\tilde{p}$ for (12) is $\tilde{p}(\mathbf{x}_1 \mid \mathbf{z}) \propto p(\mathbf{x}_1 \mid \mathbf{z}) \exp(\beta r(\mathbf{x}_1, \mathbf{z}))$ for some inverse temperature $\beta$. The marginal KL is intractable for diffusion models, so methods like DPOK [16] optimize an upper bound on the marginal KL in the form of a per-step KL penalty $-\gamma \sum_{i=1}^{T} D_{\mathrm{KL}}[\tilde{p}(\mathbf{x}_{i\Delta t} \mid \mathbf{x}_{(i-1)\Delta t}, \mathbf{z}) \| p(\mathbf{x}_{i\Delta t} \mid \mathbf{x}_{(i-1)\Delta t}, \mathbf{z})]$ added to the reward. By contrast, RTB can avoid the bias in such an approximation and directly learn to generate unbiased samples from the posterior $\tilde{p}(\mathbf{x}_1 \mid \mathbf{z})$.

**Setup.** We demonstrate how RTB can be used to fine-tune pretrained text-to-image diffusion models. We use the latent diffusion model Stable Diffusion v1-5 [63] as a prior over $512 \times 512$

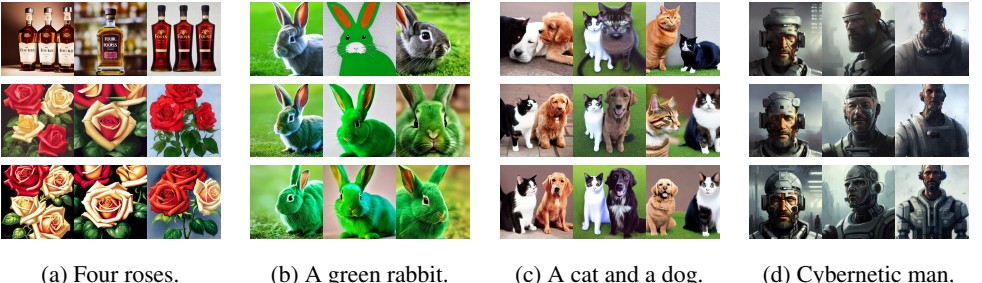

|(a) Four roses.|(b) A green rabbit.|(c) A cat and a dog.|(d) Cybernetic man.|

Figure 4: Images generated from prior (top row), DPOK (middle row) and RTB (bottom row) for 4 different prompts. Images in the same column share the random DDIM seed. More images in §H.2.

images. Following DPOK [16], we use ImageReward [87], which has been trained to match human preferences as well as prompt accuracy to attributes such as the number of objects, color, and compositionality, as the reward $\log r(\mathbf{x}_1, \mathbf{z})$. As reference, we present comparisons against DPOK with the default KL regularization $\gamma = 0.01$ and DPOK with $\gamma = 0.0$, which is equivalent to DDPO [8]. We measure the final average reward and the diversity of the generated image, as measured by the average pairwise cosine distance between CLIP embeddings [58] of a batch of generated images. Further details about the experimental setup and ablations are discussed in §H.

**Results.** Fig. 3 plots the diversity versus log reward on a set of prompts from [16, 87]. In terms of average $\log r(\mathbf{x}_1, \mathbf{z})$, RTB either matches or outperforms DPOK, while generally achieving lower reward than DDPO. The CLIP diversity score for RTB and DPOK are on average higher than DDPO, which is expected since it does not use KL regularization. For qualitative image assessments, refer to Fig. 4 and §H.2. Through this experiment, we show that RTB scales well to high dimensional, multimodal data, matching state-of-the-art methods for fine-tuning text-to-image diffusion models.

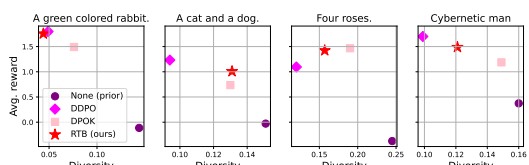

Figure 3: Fine-tuning Stable Diffusion with ImageReward. We report mean $\log r(\mathbf{x}_1, \mathbf{z})$ and diversity, measured as the mean cosine distance between CLIP embeddings for a batch of 100 generated images.[2]

### 3.3 Text infilling with discrete diffusion language models

To evaluate our approach on discrete diffusion models, we consider the problem of text infilling [91], which involves filling in missing tokens given some context tokens. While discrete diffusion models – unlike their continuous counterparts – can be challenging to train [4, 9, 49, 74], *score entropy discrete diffusion* [SEDD; 43] matches the language modeling performance of autoregressive language models of similar scale. Non-autoregressive generation in diffusion language models can provide useful inductive biases for infilling, such as the ability to attend to context on both sides of a target token.

**Setup.** We use the ROCStories corpus [50], a dataset of short stories containing 5 sentences each. We adopt the task setup from [29], where the first 3 sentences of a story $\mathbf{x}$ and the last sentence $\mathbf{y}$ are given, and the goal is to generate the fourth sentence $\mathbf{z}$ such that the overall story is coherent and consistent. The fourth sentence can involve a turning point in the story and is thus challenging to fill in. We aim to model the posterior $p^{\text{post}}(\mathbf{z} \mid \mathbf{x}, \mathbf{y}) \propto p(\mathbf{z} \mid \mathbf{x})p_{\text{reward}}(\mathbf{y} \mid \mathbf{x}, \mathbf{z})$ where $p$ is a SEDD language model prior (a conditional model over $\mathbf{z}$ given $\mathbf{x}$) and $p_{\text{reward}}$ is an autoregressive language model fine-tuned with a maximum likelihood objective on a held-out subset of the dataset. As baselines, we consider simply prompting the diffusion language model with $\mathbf{x}$ (Prompt $(\mathbf{x})$) and $\mathbf{x}, \mathbf{y}$ (Prompt $(\mathbf{x}, \mathbf{y})$). Additionally, to contextualize the performance, we also consider autoregressive language model baselines from [29], which studied this problem under an autoregressive prior $p(\mathbf{z} \mid \mathbf{x})$. SFT is trained on $50,000$ examples compared to 1000 for RTB, and serves as an upper bound on the performance in this task. See §F for further details about the experimental setup.

---

[2]Full prompt for "Cybernetic man": "A half - masked rugged laboratory engineer man with cybernetic enhancements as seen from a distance, scifi character portrait by greg rutkowski, esuthio, craig mullins."

**Results.** Following [29], we use three standard metrics to measure the similarity of the generated infills with the reference infills from the dataset: BERTScore [90] (with De-BERTa [26]), BLEU-4 [54], and GLEU-4 [86]. Table 3 summarizes the results. We observe that the diffusion language model performs significantly better than the autoregressive language model without any fine-tuning. RTB further improves the performance over prompting, and even outperforms the strongest autoregressive baseline of GFlowNet fine-tuning. We provide some examples of generated text in §F.

Table 3: Results on the story infilling task with autoregressive and discrete diffusion language models. Metrics are computed with respect to reference infills from the dataset. All metrics are mean±std over 5 samples for each of the 100 test examples. RTB with discrete diffusion prior performs better than best baseline with autoregressive prior.

| Model | Algorithm ↓ Metric → | BLEU-4 | GLEU-4 | BERTScore |
|---|---|---|---|---|
| Autoreg. | Prompting | $0.010_{\pm 0.002}$ | $0.022_{\pm 0.001}$ | $0.005_{\pm 0.001}$ |
| | Supervised fine-tuning | $0.012_{\pm 0.001}$ | $0.023_{\pm 0.001}$ | $0.013_{\pm 0.002}$ |
| | GFN fine-tuning [29] | $0.019_{\pm 0.001}$ | $0.031_{\pm 0.002}$ | $0.102_{\pm 0.005}$ |
| Discrete diffusion | Prompt $(\mathbf{x})$ | $0.011_{\pm 0.002}$ | $0.023_{\pm 0.002}$ | $0.014_{\pm 0.003}$ |
| | Prompt $(\mathbf{x}, \mathbf{y})$ | $0.014_{\pm 0.003}$ | $0.027_{\pm 0.003}$ | $0.092_{\pm 0.004}$ |
| | **RTB (ours)** | $0.025_{\pm 0.002}$ | $0.045_{\pm 0.002}$ | $0.156_{\pm 0.003}$ |
| | SFT (upper bound) | $0.031_{\pm 0.002}$ | $0.057_{\pm 0.004}$ | $0.182_{\pm 0.005}$ |

## 3.4  KL-constrained policy search in offline reinforcement learning

The goal of RL algorithms is to learn a policy $\pi(a \mid s)$, *i.e.*, a mapping from states $s$ to actions $a$ in an environment, that maximizes the expected cumulative discounted reward [75]. In the offline RL setting [39], the agent has access to a dataset $\mathcal{D} = \{(s_t^i, a_t^i, s_{t+1}^i, r_t^i)\}_{i=1}^N$ of transitions (where each sample $(s_t, a_t, s_{t+1}, r_t)$ indicates that an agent taking action $a_t$ at state $s_t$ transitioned to the next state $s_{t+1}$ and received reward $r_t$). This dataset is assumed to be generated by a *behavior policy* $\mu(a \mid s)$, which may be a diffusion model trained on $\mathcal{D}$. Offline RL algorithms must learn a new policy $\pi$ which achieves high return using only this dataset without interacting with the environment.

An important problem in offline RL is policy extraction from trained $Q$-functions [55, 25, 44]. For reliable extrapolation, one wants the policy to predict actions that have high $Q$-values, but also have high density under the behavior policy $\mu$, as naive maximization can result in choosing actions with low probability under $\mu$ and thus unreliable predictions from the $Q$-function. This is formulated as a KL-constrained policy search problem:

$$\operatorname*{argmax}_{\pi} \mathbb{E}_{s \sim d_\mu, a \sim \pi(a|s)}[Q(s,a)], \quad \mathbb{E}_{s \sim d_\mu}[D_{\mathrm{KL}}(\pi(a \mid s) \parallel \mu(a \mid s))] \leq \epsilon, \tag{13}$$

where $d_\mu$ is the distribution over states induced by following the policy $\mu$. The optimal policy $\pi$ in (13) is the product distribution $\pi^*(a \mid s) \propto \mu(a \mid s) \exp(\beta Q(s,a))$ for some inverse temperature $\beta$. If $\mu(a \mid s)$ is a conditional diffusion model over continuous actions $a$ conditioned on state $s$, we use RTB to fine-tune a diffusion behavior policy to sample from $\pi^*$, using $\mu$ as the prior and $\exp(\beta Q(s,a))$ as the target constraint. We use a $Q$-function trained using IQL [36].

**Setup.** We test on continuous control tasks in the D4RL suite [18], which consists of offline datasets collected using a mixture of SAC policies of varying performance. We evaluate on the halfcheetah, hopper and walker2d MuJoCo [76] locomotion tasks, each of which contains three datasets of transitions: "medium" (collected from an early-stopped policy), "medium-expert" (collected from both an expert and an early-stopped policy) and "medium-replay" (transitions stored in the replay buffer prior to early stopping). We compare against standard offline RL baselines (Behavior Cloning (BC), CQL [37], and IQL [36]) and diffusion-based offline RL methods which are currently state-of-the-art: Diffuser [D; 30], Decision Diffuser [DD; 2], D-QL [83], IDQL [25], and QGPO [44]. For algorithm implementation details, hyperparameters, and a report of baselines, see §G.

**Results.** Table 4 shows that RTB matches state-of-the-art results across the D4RL tasks. In particular, RTB performs strongly in the medium-replay tasks, which contain the most suboptimal data and consequently the poorest behavior prior. We highlight that our performance is similar to QGPO [44], which learns intermediate energy densities for diffusion posterior sampling.

## 4  Other related work

**Composing iterative generative processes.** Beyond the approximate posterior sampling algorithms and application-specific techniques discussed in §1 and §3, several recent works have explored the use of hierarchical models, such as diffusion models, as modular components in generative processes. Diffusion models can be used to sample product distributions to induce compositional structure in images [41, 15]. Amortized Bayesian inference [35, 60, 59, 20] is another domain of sampling from product distributions where diffusion models are now being used [21]. Beyond product models, [19]

Table 4: Average rewards of trained policies on D4RL locomotion tasks (mean±std over 5 random seeds). Following past work, numbers within 5% of maximum in every row are highlighted.

| Task ↓ Algorithm → | BC | CQL | IQL | D | DD | D-QL | IDQL | QGPO | **RTB (ours)** |
|---|---|---|---|---|---|---|---|---|---|
| halfcheetah-medium-expert | 55.2 | 91.6 | 86.7 | 79.8 | $90.6_{\pm1.3}$ | $96.1_{\pm0.3}$ | 95.9 | $93.5_{\pm0.3}$ | $74.93_{\pm1.72}$ |
| hopper-medium-expert | 52.5 | 105.4 | 91.5 | 107.2 | $111.8_{\pm1.8}$ | $110.7_{\pm1.3}$ | 108.6 | $108.0_{\pm2.5}$ | $96.71_{\pm3.53}$ |
| walker2d-medium-expert | 107.5 | 108.8 | 109.6 | 108.4 | $108.8_{\pm1.7}$ | $109.7_{\pm0.3}$ | 112.7 | $110.7_{\pm0.6}$ | $109.52_{\pm0.11}$ |
| halfcheetah-medium | 42.6 | 44.0 | 47.4 | 44.2 | $49.1_{\pm1.0}$ | $50.6_{\pm0.5}$ | 51.0 | $54.1_{\pm0.4}$ | $53.70_{\pm0.33}$ |
| hopper-medium | 52.9 | 58.5 | 66.3 | 58.5 | $79.3_{\pm3.6}$ | $82.4_{\pm4.6}$ | 65.4 | $98.0_{\pm2.6}$ | $82.76_{\pm7.07}$ |
| walker2d-medium | 75.3 | 72.5 | 78.3 | 79.7 | $82.5_{\pm1.4}$ | $85.1_{\pm0.9}$ | 82.5 | $86.0_{\pm0.7}$ | $87.29_{\pm3.15}$ |
| halfcheetah-medium-replay | 36.6 | 45.5 | 44.2 | 42.2 | $39.3_{\pm4.1}$ | $47.5_{\pm0.3}$ | 45.8 | $47.6_{\pm1.4}$ | $48.11_{\pm0.56}$ |
| hopper-medium-replay | 18.1 | 95.0 | 94.7 | 96.8 | $100.0_{\pm0.7}$ | $100.7_{\pm0.6}$ | 92.1 | $96.9_{\pm2.6}$ | $100.40_{\pm0.21}$ |
| walker2d-medium-replay | 26.0 | 77.2 | 73.9 | 61.2 | $75.0_{\pm4.3}$ | $94.3_{\pm1.5}$ | 85.1 | $84.4_{\pm4.1}$ | $93.57_{\pm2.63}$ |

studies ways to amortize other kinds of compositions of hierarchical processes, including diffusion models, while [67] proposes methods to sample the product of many iterative processes in application to federated learning. Finally, models without hierarchical structure, such as normalizing flows, have been used to amortize intractable inference in pretrained diffusion models [*e.g.*, 17]. In contrast, our method performs posterior inference by *fine-tuning* a prior model, developing a direction on flexible extraction of information from large pretrained models [29].

**Diffusion samplers.** Several prior works seek to amortize MCMC sampling from unnormalized densities by training diffusion models for efficient mode-mixing [6, 89, 78, 62, 79, 3]. Our work is most closely related to continuous GFlowNets [38], which offer an alternative perspective on training diffusion samplers using off-policy flow consistency objectives [38, 88, 65]. Recently, Berner et al. [7] have shown connections among existing families of diffusion sampling algorithms and analyzed their continuous-time limits.

# 5 Conclusions and future work

Relative trajectory balance provides a new approach to training diffusion models to generate unbiased posterior samples given a diffusion prior and an arbitrary reward function. Through experiments on a variety of domains – vision, language, continuous control – we demonstrated the flexibility and general applicability of RTB. RTB can be optimized with off-policy trajectories, and future work can explore ways to leverage off-policy training, using techniques such as local search [34, 65] to improve sample efficiency and mode coverage. Simulation-based objectives in the style of [89] are also applicable to the amortized sampling problems we consider and should be explored, as should simulation-free extensions, *e.g.*, through objectives that are local in time [45]. The ability to handle arbitrary black-box likelihoods also makes RTB a useful candidate for inverse problems in domains such as 3D object synthesis with likelihood computed via a renderer [*e.g.*, 56, 82], imaging problems in astronomy [*e.g.*, 1], medical imaging [*e.g.*, 73], and molecular structure prediction [*e.g.*, 84].

Moreover, RTB could facilitate a breakthrough in modeling molecular dynamics—a notoriously challenging task due to the need to sample rare-event trajectories in chemical simulations—by converting these problems into posterior inference over amplified distributions of rare-event samples. Notably, Seong et al. [66] have already explored a preliminary version of this concept by employing TB with a reward multiplied by the prior likelihood, which is effectively equivalent to RTB.

**Limitations.** RTB learns the posterior through simulation-based training, which can be slow and memory-intensive. Additionally, the RTB objective is computed on complete trajectories without any local credit-assignment signal, which can result in high variance in the gradients. Guarantees on the error incurred by imperfect fit of the prior model, amortization, and time discretization (analogous to [7]'s analysis for diffusion samplers) have not been obtained and should be considered in future work.

**Broader impact.** While our contributions focus on an algorithmic approach for learning posterior samplers with diffusion priors, we acknowledge that like other advances in generative modelling, our approach can potentially be used by nefarious actors to train generative models to produce harmful content and misinformation. At the same time, our approach can be also be used to mitigate biases captured in pretrained models and applied to various scientific problems.

## Acknowledgments and Disclosure of Funding

The authors thank Adam Coogan, Yashar Hezaveh, Guillaume Lajoie, and Laurence Perreault Levasseur for helpful suggestions in the course of this project and Mandana Samiei for comments on a draft of the paper.

The authors acknowledge funding from CIFAR, NSERC, IVADO, UNIQUE, FACS Acuité, NRC AI4Discovery, Samsung, and Recursion.

The research was enabled in part by computational resources provided by the Digital Research Alliance of Canada (`https://alliancecan.ca`), Mila (`https://mila.quebec`), and NVIDIA.

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

## A  Proofs

**Proposition 1** (Relative TB constraint). *If $p_\theta$, $p_\phi^{\text{post}}$, and the scalar $Z_\phi$ jointly satisfy the* relative *trajectory balance (RTB) constraint*

$$Z_\phi \cdot p_\phi^{\text{post}}(\mathbf{x}_0, \mathbf{x}_{\Delta t}, \dots, \mathbf{x}_1) = r(\mathbf{x}_1) p_\theta(\mathbf{x}_0, \mathbf{x}_{\Delta t}, \dots, \mathbf{x}_1) \tag{8}$$

*for every denoising trajectory $\mathbf{x}_0 \to \mathbf{x}_{\Delta t} \to \cdots \to \mathbf{x}_1$, then $p_\phi^{\text{post}}(\mathbf{x}_1) \propto p_\theta(\mathbf{x}_1) r(\mathbf{x}_1)$, i.e., the diffusion model $p_\phi^{\text{post}}$ samples the posterior distribution. Furthermore, if $p_\theta$ also satisfies the TB constraint (7) with respect to the noising process $q$ and some target density $q(\mathbf{x}_1)$, then $p_\phi^{\text{post}}$ satisfies the TB constraint with respect to the target density $q^{\text{post}}(\mathbf{x}_1) \propto q(\mathbf{x}_1) r(\mathbf{x}_1)$, and $Z = \int q(\mathbf{x}_1) r(\mathbf{x}_1) \, d\mathbf{x}_1$.*

*Proof of Prop. 1.* Suppose that $p_\theta$, $p_\phi^{\text{post}}$, and $Z$ jointly satisfy (8). Then necessarily $Z \neq 0$, since the quantities on the right side are positive. We then have, using (4),

$$\begin{aligned}
p_\phi^{\text{post}}(\mathbf{x}_1) &= \int p_\phi^{\text{post}}(\mathbf{x}_0, \mathbf{x}_{\Delta t}, \dots, \mathbf{x}_1) \, d\mathbf{x}_0 \, d\mathbf{x}_{\Delta t} \dots d\mathbf{x}_{1-\Delta t} \\
&= \frac{1}{Z} r(\mathbf{x}_1) \int p_\theta(\mathbf{x}_0, \mathbf{x}_{\Delta t}, \dots, \mathbf{x}_1) \, d\mathbf{x}_0 \, d\mathbf{x}_{\Delta t} \dots d\mathbf{x}_{1-\Delta t} \\
&= \frac{1}{Z} r(\mathbf{x}_1) p_\theta(\mathbf{x}_1) \qquad\qquad\qquad\qquad\qquad \propto p_\theta(\mathbf{x}_1) r(\mathbf{x}_1),
\end{aligned}$$

as desired.

Now suppose that $p_\theta$ also satisfies the TB constraint (7) with respect to $q(\mathbf{x}_1)$. Then, for any denoising trajectory,

$$q(\mathbf{x}_0, \mathbf{x}_{\Delta t}, \dots, \mathbf{x}_{1-\Delta t} \mid \mathbf{x}_1) = \frac{p_\theta(\mathbf{x}_0, \mathbf{x}_{\Delta t}, \dots, \mathbf{x}_1)}{q(\mathbf{x}_1)} = \frac{p_\phi^{\text{post}}(\mathbf{x}_0, \mathbf{x}_{\Delta t}, \dots, \mathbf{x}_1)}{q(\mathbf{x}_1) r(\mathbf{x}_1)/Z}. \tag{14}$$

showing that $p_\phi^{\text{post}}$ satisfies the TB constraint with respect to the noising process $q$ and the (not yet shown to be normalized) density $\frac{1}{Z} q(\mathbf{x}_1) r(\mathbf{x}_1)$. We integrate out the variables $\mathbf{x}_0, \mathbf{x}_{\Delta t}, \dots, \mathbf{x}_{1-\Delta t}$ in (14), giving

$$1 = \frac{p_\phi^{\text{post}}(\mathbf{x}_1)}{q(\mathbf{x}_1) r(\mathbf{x}_1)/Z}$$
$$q(\mathbf{x}_1) r(\mathbf{x}_1) = Z p_\phi^{\text{post}}(\mathbf{x}_1).$$

Integrating over $\mathbf{x}_1$ shows $\int q(\mathbf{x}_1) r(\mathbf{x}_1) \, d\mathbf{x}_1 = Z$. $\qquad\qquad\qquad\qquad\qquad\square$

## B  Relative TB as TB under the prior measure

The theoretical foundations for continuous generative flow networks [38] establish the correctness of enforcing constraints such as trajectory balance (7) for training sequential samplers, such as diffusion models, to match unnormalized target densities. While we have considered Gaussian transitions and identified transition kernels with their densities with respect to the Lebesgue measure over $\mathbb{R}^d$, these foundations generalize to more general *reference measures*. In application to diffusion samplers, suppose that $\pi_{\text{ref}}(\mathbf{x}_t)$ is a collection of Lebesgue-absolutely continuous densities over $\mathbb{R}^d$ for $t = 0, \Delta t, \dots, 1$ and that $\overrightarrow{\pi}_{\text{ref}}(\mathbf{x}_t \mid \mathbf{x}_{t-\Delta t})$, $\overleftarrow{\pi}_{\text{ref}}(\mathbf{x}_{t-\Delta t} \mid \mathbf{x}_t)$ are collections of Lebesgue-absolutely continuous transition kernels. If these densities jointly satisfy the detailed balance condition $\pi_{\text{ref}}(\mathbf{x}_t) \overleftarrow{\pi}_{\text{ref}}(\mathbf{x}_{t-\Delta t} \mid \mathbf{x}_t) = \pi_{\text{ref}}(\mathbf{x}_{t-\Delta t}) \overrightarrow{\pi}_{\text{ref}}(\mathbf{x}_t \mid \mathbf{x}_{t-\Delta t})$, then they satisfy the conditions to be reference measures. A main result of [38] is that if a pair of forward and backward processes satisfies the trajectory balance constraint (7) jointly with a reward density $r$, then the forward process $p$ samples from the distribution with density $r$, with all densities interpreted as *relative to the reference measures* $\pi_{\text{ref}}, \overleftarrow{\pi}_{\text{ref}}, \overrightarrow{\pi}_{\text{ref}}$.[3]

---

[3] Recall that the relative density (or Radon-Nikodym derivative) of a distribution with density $p$ under the Lebesgue measure relative to one with density $\pi$ is simply the ratio of densities $p/\pi$.

If $p_\theta$ is a diffusion model that satisfies the TB constraint jointly with some reverse process $q$ and target density $q(\mathbf{x}_1)$, then one can take the reference transition kernels $\overrightarrow{\pi}_{\text{ref}}, \overleftarrow{\pi}_{\text{ref}}$ to be $p$ and $q$, respectively. In this case, the TB constraint for a target density $\frac{1}{Z}r(\mathbf{x}_1)$ and forward transition $p_\phi^{\text{post}}$ is

$$\frac{p_\phi^{\text{post}}(\mathbf{x}_0, \mathbf{x}_{\Delta t}, \ldots, \mathbf{x}_1)}{\overrightarrow{\pi}_{\text{ref}}(\mathbf{x}_0, \mathbf{x}_{\Delta t}, \ldots, \mathbf{x}_1)} = \frac{\frac{1}{Z}r(\mathbf{x}_1)q(\mathbf{x}_0, \mathbf{x}_{\Delta t}, \ldots, \mathbf{x}_{1-\Delta t} \mid \mathbf{x}_1)}{\overleftarrow{\pi}_{\text{ref}}(\mathbf{x}_0, \mathbf{x}_{\Delta t}, \ldots, \mathbf{x}_{1-\Delta t} \mid \mathbf{x}_1)}, \tag{15}$$

which is identical to the RTB constraint (8). If (15) holds, then $p_\phi^{\text{post}}$ samples from the distribution with density $\frac{1}{Z}r(\mathbf{x}_1)$ relative to $\pi_{\text{ref}}(\mathbf{x}_1)$, which is exactly $\frac{1}{Z}p_\theta(\mathbf{x}_1)r(\mathbf{x}_1)$. We have thus recovered RTB as a case of TB for non-Lebesgue reference measures.

## C Posterior inference on two-dimensional Gaussian mixture model

**Setup** We conduct toy experiments in low-dimensional spaces using samples from a Gaussian mixture model with multiple modes to visually demonstrate its validity. The prior distribution $p(\mathbf{x}_1)$ is trained on a Gaussian mixture model with 25 evenly weighted modes, while the target posterior $p^{\text{post}}(\mathbf{x}_1) = r(\mathbf{x}_1)p(\mathbf{x}_1)$ uses a reward $r(\mathbf{x}_1)$ to select and re-weight 9 modes from $p(\mathbf{x}_1)$. More specifically, the resulting posterior is:

$$p^{\text{post}}(\mathbf{x}_1) = \frac{1}{\sum_j \tilde{\pi}_j} \sum_i \tilde{\pi}_i \mathcal{N}(\mathbf{x}_1 \mid \mu_i, \mathbf{I}) \tag{16}$$

$$\{\mu_i\} = \{(-10, -5), (-5, -10), (-5, 0), (10, -5), (0, 0), (0, 5), (5, -5), (5, 0), (5, 10)\} \tag{17}$$

$$\{\tilde{\pi}_i\} = \{4, 10, 4, 5, 10, 5, 4, 15, 4\} \tag{18}$$

Our objective is to sample from the posterior $p^{\text{post}}(\mathbf{x}_1)$. We compare our method with several baselines, including policy gradient reinforcement learning (RL) with KL constraint and classifier-guided diffusion models. For RL, we implemented the REINFORCE method with a mean baseline and a KL constraint, following recent work training diffusion models to optimize a reward function [8]. Sampling according to the RL policy leads to a distribution $q_\theta(\mathbf{x}_1)$, which is trained with the objective:

$$J(\theta) = \mathbb{E}_{q_\theta(\mathbf{x}_1)}[r(\mathbf{x}_1)] + \alpha D_{\text{KL}}(q_\theta(\mathbf{x}_1)\|p(\mathbf{x}_1)) \tag{19}$$

While the exact computation of $KL(q_\theta(\mathbf{x}_1)\|p(\mathbf{x}_1))$ is intractable, we follow the approximation method introduced by Fan et al. [16], which sums the divergence at every diffusion step. This approximation optimizes an upper bound of the marginal KL.

The other baseline is classifier (energy) guidance, which given a diffusion prior, samples using a posterior score function estimate:

$$\nabla_{\mathbf{x}_t} \log p^{\text{post}}(\mathbf{x}_t) \approx \nabla_{\mathbf{x}_t} \log p(\mathbf{x}_t) + \nabla_{\mathbf{x}_t} \log r(\mathbf{x}_t) \tag{20}$$

Note that this is a biased approximation of the true intractable score:

$$\nabla_{\mathbf{x}_t} \log p^{\text{post}}(\mathbf{x}_t) = \nabla_{\mathbf{x}_t} \log p(\mathbf{x}_t) + \nabla_{\mathbf{x}_t} \log \mathbb{E}_{p(\mathbf{x}_1|\mathbf{x}_t)}[r(\mathbf{x}_1)] \tag{21}$$

For our experiments, we follow the source code[4] provided in recent diffusion sampler benchmarks [65]. We utilize a batch size of 500, with finetuning at 5,000 training iterations, a learning rate of 0.0001, a diffusion time scale of 5.0, 100 steps, and a log variance range of 4.0. The neural architecture employed is identical to that used in [65]. For pretraining the prior model, we use the same hyperparameters as above, but with 10,000 training iterations using maximum likelihood estimation with true samples.

**Results.** As we reported in the main text, in Fig. 1, we present illustrative results. The classifier-guided diffusion model shows biased posterior sampling (Fig. 1f), failing to provide accurate inference. RL with a per step KL constraint cannot exactly optimize for the posterior distribution, making the tuning of the KL weight $\alpha$ crucial to achieving desirable output Fig. C.1. RTB asymptotically achieves the true posterior without introducing a balancing hyperparameter $\alpha$. Another advantage of our approach is off-policy exploration for efficient mode coverage. RL methods for fine-tuning diffusion models (e.g., DPOK [16], DDPO [8]) typically use policy gradient style methods that are on-policy. By using a simple off-policy trick introduced by [47, 38] and demonstrated by Sendera et al. [65], we can introduce randomness into the exploration process in diffusion by adding $\frac{\epsilon^2}{T}$, where $\epsilon$ is a noise hyperparameter and $T$ is the diffusion timestep, into the variances and annealing it to zero over training iterations. We set $\epsilon = 0.5$ for off-policy exploration. As shown in Fig. C.2, RTB with off-policy exploration gives very close posterior inferences, whereas off-policy exploration in RL with $\alpha = 0.5$ (which is a carefully selected hyperparameter) does not improve performance due to its on-policy nature.

## D Code

Code for all experiments is available at https://github.com/GFNOrg/diffusion-finetuning and will continue to be maintained and extended.

---

[4] https://github.com/GFNOrg/gfn-diffusion

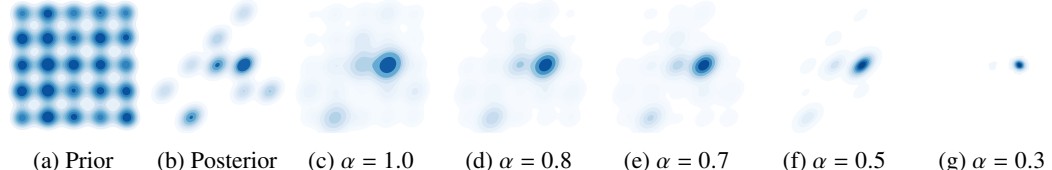

(a) Prior     (b) Posterior     (c) $\alpha = 1.0$     (d) $\alpha = 0.8$     (e) $\alpha = 0.7$     (f) $\alpha = 0.5$     (g) $\alpha = 0.3$

Figure C.1: Tuning the KL weight $\alpha$ in reinforcement learning: influences the balance between sticking to the prior distribution and moving towards the modes of the reward density. A higher $\alpha$ value maintains closer adherence to the prior, while a lower $\alpha$ allows a gradual shift towards high values of $r(\mathbf{x})$. Setting $\alpha$ below 0.3 tends to cause mode collapse, moving too far from the prior and focusing on maximizing rewards for single modes. $\alpha = 0.5$ gives us samples that closest resembles the posterior.

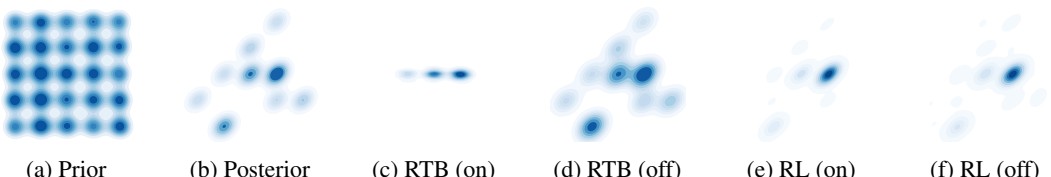

(a) Prior     (b) Posterior     (c) RTB (on)     (d) RTB (off)     (e) RL (on)     (f) RL (off)

Figure C.2: Off-policy exploration benefits for RTB training. RTB, with simple off-policy exploration techniques that increase randomness in the diffusion process, significantly improves mode coverage. On the other hand, policy gradient RL methods which are typically used to finetune diffusion models are on-policy, and hence prone to mode collapse.

# E    On classifier guidance and RTB posterior sampling

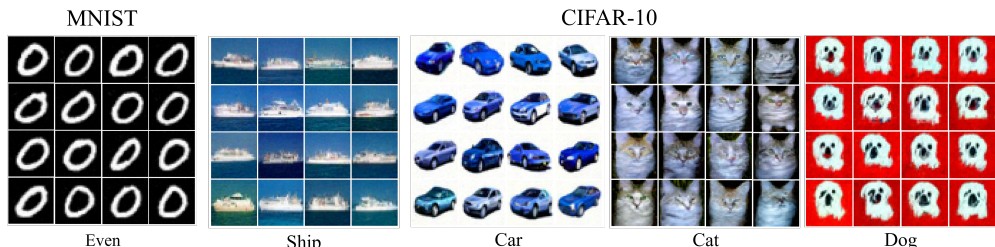

Figure E.1: Samples from a posterior model fine-tuned with RL (no KL). We observe early mode collapse, showcasing high-reward samples with minimal diversity.

## E.1    Experimental Details

In our experiments, we fine-tune pretrained unconditional diffusion models with our RTB objective, to sample from a posterior distribution in the form $p^{\text{post}}(x \mid y) = p(y \mid x)p(x)$. In this section, we detail the experimental settings for RTB as well as the compared baselines.

**Experiments setting.** For MNIST, we pretrain a noise-predicting diffusion model on $28 \times 28$ (upscaled to $32 \times 32$) single channel images of digits from the MNIST datasets. We discretize the forward and backward processes into 200 steps and train our model until convergence. For CIFAR-10, we use a pretrained model from [27], trained to generate $32 \times 32$ 3-channel images from the CIFAR-10 dataset, while discretizing the noising/denoising processes into 1000 steps. For fine-tuning the prior, we parametrize the posterior with LoRA weights [28], with the number of parameters equal to about 3% of the prior model's parameter count. We train our models on a single NVIDA V100 GPU.

We compute FID as a similarity score estimate of the *true* posterior distribution from the data. As such, the computation is limited to the total number of per-class-samples present in the data, (between 5k and 6k for CIFAR-10 and MNIST digits, and 30k for the even/odd task).

**RTB.** For RTB fine-tuning, we finetune a diffusion model following the objective in Equation 9. We impose the objective while sampling denoising paths following a DDPM sampling scheme, with only 20% to 50% of the original trained steps. We employ loss clipping at 0.1, to account for imperfect constraints in the pretrained prior, and train each of our models for 1500 training iterations, well into convergence trends.

**RL [16].** We implement two RL-based fine-tuning techniques derived from DPOK [16] and DDPO [8], respectively with and without KL regularization. These implementations use a reinforcement learning baseline similar to the one in our experiments described in §3.2. By following the same sampling scheme as in our RTB experiments, we enable a direct comparison with RTB. To fine-tune the KL weight, we perform a search over $\alpha \in \{0.01, 0.1, 1.0\}$.

**DP [11].** We implement and adapt the Gaussian version of the posterior sampling scheme in [11], originally devised for noisy inverse problems. This method relaxes some of our experimental constraints, as it requires a differentiable reward $r(\mathbf{x})$. We perform a sweep over ten values of the suggest parameter range for the step size $\zeta \in [.1, 1.]$ on MNIST single-digit sampling, and choose $\zeta = 0.1$ for our experiments.

**LGD-MC [70].** We adapt the implementation of the algorithm in [70] to sample from the classifier-based posteriors in CIFAR-10 and MNIST. Similarly to the DP baseline, we use our pretrained classifier to perform measurements at each sampling step, and use a Monte Carlo estimate of the gradient correction to guide the denoising process. We choose $\zeta = 0.1$ following the DP experiments and default the number of particles to 10 as per the authors' guidelines.

### E.2 Additional findings.

**Classifier-guidance baselines.** We find that the DP and LGD-MC classifier-guidance based baselines struggle to sample from the true posterior distribution in our experimental settings. The baselines achieve the lowers classifier average rewards in all tested settings. Despite choosing $\zeta = 0.1$ as the validate best performing hyperparameter, we also also observe the posterior samples from DP and LGD-MC to be close to the prior. As such, DP and LGD-MC score high in diversity, and low in FID for the Even/Odd experimental scenario, as expected from prior sampling benchmarks, but failing to appropriately model the posterior distribution.

**RL and mode collapse.** In the pure Reinforcement Learning objective imposed for the experiments in §3.1 (no KL), we observe a significantly higher reward than other baseline methods, while showcasing increased FID and lower diversity. In Fig. E.1 we show a random set of 16 samples for posterior models trained on 4 different classes of the CIFAR-10 datasets, as well as the *Even* objective from the MNIST dataset, after 500 training iterations. In the figure, we observe early mode collapse and reward exploitation, visually evident from the little to no variation amongst samples for each class class, and single-digit collapse in the multi-modal *even* digits objective (see samples in Fig. 2 for comparison with our RTB-finetuned models).

**Conditional architectures.** We repeat the even/odd posterior sampling experiment of §3.1 in a conditional setting, where the condition is an input to the posterior model. For the posterior architecture, we use a naive modification of the prior with an extra input channel, which is populated a full mask of 0 or 1 for conditioning on the even and odd classes, respectively. The results are shown in Table E.1. We look forward to future work which develops more specialized architectures for handling conditional constraints.

Table E.1: Conditional experiment for MNIST even/odd posterior. Note that the posterior model in the conditional experiment is different from that in the baselines because it uses a different architecture that includes an additional input channel.

| Dataset → | MNIST even/odd | |
|---|---|---|
| Algorithm ↓ Metric → | $\mathbb{E}[\log r(\mathbf{x})]$ (↑) | FID (↓) |
| DPS | $-1.2270_{\pm 0.202}$ | $1.1498_{\pm 0.182}$ |
| LGD−MC | $-1.1720_{\pm 0.199}$ | $1.1445_{\pm 0.184}$ |
| DDPO | $-8.6_{\pm 12.3} \times 10^{-11}$ | $1.8024_{\pm 0.423}$ |
| DPOK | $-0.0783_{\pm 0.082}$ | $1.2536_{\pm 0.206}$ |
| **RTB (unconditional)** | $-0.1816_{\pm 0.175}$ | $1.1794_{\pm 0.171}$ |
| **RTB (conditional)** | -0.1236 | 0.9112 |

## F    Infilling with discrete diffusion

**Additional details.** We illustrate some examples from the ROC Stories dataset used for training in Table F.1. For the prior we use the `sedd-small`[5] model, which uses an absorbing noising process [4] with a log linear noise schedule, as the diffusion prior $p(\mathbf{z} \mid \mathbf{x})$. The posterior model is

---

[5]<https://huggingface.co/louaaron/sedd-small>

Table F.1: Examples of training samples for the language infilling task.

| Beginning (x) | Middle (z) | End (y) |
|---|---|---|
| I was going to a Halloween party. I looked through my clothes but could not find a costume. I cut up my old clothes and constructed a costume. | I put my costume on and went to the party. | My friends loved my costume. |
| Allen thought he was a very talented poet. He attended college to study creative writing. In college, he met a boy named Carl. | Carl told him that he wasn't very good. | Because of this, Allen swore off poetry forever. |

parameterized as a copy of the prior. To condition the diffusion model on the beginning **x** we set the tokens at the appropriate location in the state in the initial time step, *i.e.* $t = 0$. Our implementation is based on the original SEDD codebase [6]. Training this model is computationally expensive (in terms of memory and speed) so we utilize the stochastic TB trick, only propagating the gradients through a subset of the steps of the trajectory. We also use the loss clipping trick as discussed in §2.3. Specifically, we clip the loss below a certain threshold to 0, resulting in updates only when the loss is larger. This threshold – referred to as the loss clipping coefficient – is a hyperparameter. As this is a conditional problem we also use the relative VarGrad objective. We also use some tempering on the reward likelihood which helps in learning (*i.e.*, $p_{\text{reward}}(\mathbf{y} \mid \mathbf{x}, \mathbf{z})^{\beta}$) where $\beta$ is the inverse temperature parameter. We perform all experiments on an NVIDIA A100-Large GPU. Note that we also tried a baseline of simply fine-tuning the diffusion model on the data but encountered some training instabilities that we could not fix. The hyperparameters used for training RTB in our experiments are detailed in Table F.2.

**Reward.** For training $p_{\text{reward}}$ we follow the training procedure and implementation from [29][7]. Specifically, we fine-tune a `GPT-2 Large` model [57] on the stories dataset with full parameter fine-tuning using the `trl` library [81]. We trained for 20 epochs with a batch size of 64 and 32 gradient accumulation steps and a learning rate of 0.0005.

**Baselines.** For the baselines, we adopt the implementations from [29]. A critical difference in our experiments compared to [29] is that the posterior model is not initialized with a base model that is fine-tuned on the stories dataset. To condition the model on $X$ and $Y$, as well as for the prompting baseline, we use the following prompt:

`"Beginning: {X}\n End: {Y}\n Middle: "`

During training for the autoregressive GFlowNet fine-tuning, a $(\mathbf{x}, \mathbf{y})$ pair is sampled from the dataset and then sample (batch size) **x**s for every $(X, Y)$, and $p_{\text{reward}}(XZY)$ is used as the reward. Both the GFlowNet fine-tuning and supervised fine-tuning baseline use LoRA fine-tuning. We use the default hyperparamteres from [29]. At test time, we sample 100 infills for each example in the test set from all the models at temperature 0.9, and average over 5 such draws.

**Additional results.** Table F.3, Table F.4 and Table F.5 illustrates some examples of the infills generated by the diffusion models. We note that the general quality of the samples is poor, due to a relatively weak prior. At the same time we can observe that the prompting baselines often generate infills that are unrelated to the current story. We also note that the RTB fine-tuned model can sometimes generate repitions as the reward model tends to assign high likelihood to repitions [85]. We also attempted a LLMEval [42] for evaluating the coherence of the stories but did not obtain statistically significant results.

---

[6] https://github.com/louaaron/Score-Entropy-Discrete-Diffusion
[7] https://github.com/GFNOrg/gfn-lm-tuning

Table F.2: Hyperparameters for the story infilling task.

| | |
|---|---|
| Batch size | 16 |
| Gradient accumulation steps | 8 |
| Learning rate | 1e-5 |
| Warmup Step | 20 |
| Optimizer | AdamW |
| Reward temperature start | 1.2 |
| Reward inverse temperature end | 0.9 |
| Reward inverse temperature horizon | 5000 |
| Number of training steps | 1500 |
| Loss clipping coefficient | 0.1 |
| Discretization steps $T$ | 15 |

Table F.3: Examples of infills generated by the posterior trained with RTB along with **reference infills** for the stories infilling task.

| Beginning (x) | Middle (z) | End (y) |
|---|---|---|
| David noticed he had put on a lot of weight recently. He examined his habits to try and figure out the reason. He realized he'd been eating too much fast food lately. | **He stopped going to burger places and started a vegetarian diet.** He reviewed his habits to try to figure out how to change He asked he thought try to cut down on the amount amount. He examined his habits to try and figure out the reason. He realized he had been eating too much fast food recently. | After a few weeks, he started to feel much better. |
| Robbie was competing in a cross country meet. He was halfway through when his leg cramped up. Robbie wasn't sure he could go on. | **He stopped for a minute and stretched his bad leg.** Robbie was sure he could go on. Robbie was sure. He was floating and twisting his leg in half then. His body just caught up with his legs. Robbie was. He held his leg forward as he went through and his | Robbie began to run again and finished the race in second place. |

# G  Offline RL

## G.1  Training details

Our method requires first training a diffusion-based behavior policy $\pi_\theta$ and a Q-function $Q_\psi$. Once $\pi_\theta$ and $Q_\psi$ are trained, The posterior policy $\pi_\gamma$ is trained using RTB, with its weights initialized to the trained behavior policy weights $\theta$.

The behavior policy $\pi_\theta$ is parametrized as a state-conditioned noise-predicting denoising diffusion probabilistic model (DDPM) [27] with a linear schedule, and 75 denoising steps. The diffusion model takes as input a state $s$, a noised action $a_t$ and a noise level $t$ and predicts the source noise $\epsilon$. The state $s$ and noised action $a_t$ are concatenated with Fourier features computed on the noise level $t$, which are then fed through a 3-layer MLP of hidden dimensionality 256, with layer normalization and a GeLU activation after each hidden layer. The behavior policy is trained using the Adam optimizer with batch size 512 and learning rate 5e-4 for 10000 epochs. The Q-function $Q_\psi$ is trained using IQL. We use the same IQL experimental configurations and training hyperparameters as in [36]. That is, we set $\tau = 0.7$. The architecture for $Q_\psi$ is a 3-layer MLP with hidden dimensionality 256 and ReLU activations, which is trained using the Adam optimizer with a learning rate 3e-4 and batch size 256 for 750000 gradient steps. The task rewards are normalized as in [36] and the target network is updated

Table F.4: Examples of infills generated by Prompt $(\mathbf{x}, \mathbf{y})$ along with **reference infills** for the stories infilling task.

| Beginning ($\mathbf{x}$) | Middle ($\mathbf{z}$) | End ($\mathbf{y}$) |
|---|---|---|
| David noticed he had put on a lot of weight recently. He examined his habits to try and figure out the reason. He realized he'd been eating too much fast food lately. | **He stopped going to burger places and started a vegetarian diet.** He'd had less opportunities to eat properly all of last. Doctors made the note of the situation. He was treated. He told him the guy for a mic replacement.\n\n He felt empty for one reason and new fresh, too. | After a few weeks, he started to feel much better. |
| Robbie was competing in a cross country meet. He was halfway through when his leg cramped up. Robbie wasn't sure he could go on. | **He stopped for a minute and stretched his bad leg.** Robbie wasn't sure Robbie's fuel tank was full. Robbie took a photograph with a close friend.\n\n Only Stacey Ebers and Rand were out there. Robbie got bigger as the position got better.\n\n | Robbie began to run again and finished the race in second place. |

Table F.5: Examples of infills generated by Prompt ($\mathbf{x}$) along with **reference infills** for the stories infilling task.

| Beginning ($\mathbf{x}$) | Middle ($\mathbf{z}$) | End ($\mathbf{y}$) |
|---|---|---|
| David noticed he had put on a lot of weight recently. He examined his habits to try and figure out the reason. He realized he'd been eating too much fast food lately. | **He stopped going to burger places and started a vegetarian diet.** David, "All I had told eat was a problem. He got the backside what about that and he made the, He made just good of fast food and spliced it down. He explained everything to them, reached them out, the problem. | After a few weeks, he started to feel much better. |
| Robbie was competing in a cross country meet. He was halfway through when his leg cramped up. Robbie wasn't sure he could go on. | **He stopped for a minute and stretched his bad leg.** Robbie and Robbie was piling. Robbie and Robbie fistfight. I said goodbye. Robbie at dinner. Robbie agreed with. He cut away a little to Robbie's pace fleetingly. He held off all the police and place. Robbie. | Robbie began to run again and finished the race in second place. |

Table G.1: Mixed vs. online training on DR4L Tasks. We report mean±std over 5 random seeds.

| Task | RTB (Online) | RTB (Mixed) |
|------|-----------|-----------|
| halfcheetah-medium-replay | 46.88±0.51 | 48.11±0.56 |
| hopper-medium-replay | 99.23±3.22 | 100.40±0.21 |
| walker2d-medium-replay | 94.01±0.28 | 93.57±2.63 |

Table G.2: Temperature $\alpha = \frac{1}{\beta}$ for D4RL tasks

| Task | $\alpha$ |
|------|------|
| halfcheetah-medium-expert | 0.1 |
| hopper-medium-expert | 0.5 |
| walker2d-medium-expert | 0.1 |
| halfcheetah-medium | 0.05 |
| hopper-medium | 0.1 |
| walker2d-medium | 0.05 |
| halfcheetah-medium-replay | 0.05 |
| hopper-medium-replay | 0.05 |
| walker2d-medium-replay | 0.1 |

with soft updates of $m = 0.005$. The posterior policy $\pi_\gamma$ is trained using the relative trajectory balance objective. $\pi_\gamma$ is also parametrized as a state-conditioned noise-predicting DDPM, initialized as a copy of the prior. We additionally use the Langevin dynamics inductive bias (11), and learn an additional MLP for the energy scaling network. The posterior noise prediction network also outputs an additive correction to the output of the prior noise prediction network. That is, the predicted noise of the posterior diffusion model is defined as $\epsilon(s, a_t, t) := \epsilon(s, a_t, t; \theta) + \epsilon(s, a_t, t; \gamma)$, where $\epsilon(\cdot; \theta)$ is the output of the prior noise prediction network and $\epsilon(\cdot; \gamma)$ is the output of the posterior noise prediction network. We train all models on a single NVIDIA A100-Large GPU. The only hyperparameter tuned per task is the temperature $\alpha$ which we show in Table G.2.

Note that in these experiments both the prior and constraint are conditioned on the state $\mathbf{s}$. To prevent having to learn a neural network for $\log Z_\phi(\mathbf{s})$, we employ a variant of VarGrad objective [61]. For each state $\mathbf{s}$ sampled in the minibatch, we further generate $k = 64$ on-policy trajectories $\tau^{(i)}{}_{i=1}^{k}$ with $\pi_\gamma$. Each of these trajectories can be used to implicitly estimate $\log Z(\mathbf{s})$:

$$\log \hat{Z}(\mathbf{s})^{(i)} = \log \pi_\theta(\tau^{(i)} \mid \mathbf{s}) + Q_\psi(\mathbf{s}, \mathbf{a}_1^{(i)}) - \log \pi_\gamma(\tau^{(i)} \mid \mathbf{s}) \tag{22}$$

We then minimize the sample variance across the batch:

$$\mathcal{L}_{\text{RTB}}^{\text{VarGrad}}(\gamma) = \frac{1}{k} \sum_{i=1}^{k} \left( \log \hat{Z}(\mathbf{s})^{(i)} - \frac{1}{k} \sum_{j=1}^{k} \log \hat{Z}(\mathbf{s})^{(j)} \right)^2 \tag{23}$$

RTB allows off-policy training so we are not restricted to train with samples generated on-policy. We thus also leverage the offline dataset, which are samples from the prior and noise them with the DDPM noising process to generate off-policy trajectories with high density under the prior. Since there are actions in the replay buffer from high reward episodes in the tasks, this can help training efficiency compared to purely online training. We ran 5 seeds of training each with mixed training (off-policy and on-policy) and pure on-policy training on the medium-replay tasks, with results shown in Table G.1, where mixed training outperforms pure online training on two of the three tasks.

### G.2 Baseline details

As is standard in offline RL, we use the reported performance numbers from the previous papers. CQL, IQL are reported from the IQL paper. Diffuser (D), DD, D-QL and QGPO are reported from the QGPO paper. Their implementation improved the performance of D and D-QL compared to their original papers. IDQL results are reported from the IDQL paper. We follow the evaluation protocol of previous work, and report the mean performance over 10 episodes, averaged across 5 random seeds at the end of training (150k training steps).

## H  Fine-tuning text-to-image diffusion models

We build off the DPOK implementation[8], which fine-tunes stable-diffusion-v1-5 with ImageReward function. The posterior model to be fine-tuned is initialized as a copy of the prior model. We use LoRA [28] since it is significantly more efficient than fine-tuning the entire model. Sampling of images is done with 50 steps of DDIM [69]. Even with LoRA, it is still difficult to fit gradients of all steps in the diffusion trajectory in memory. To help with this, we use a "stochastic subsampling" trick (§H.1).

We train all models on a single NVIDIA A100-Large GPU. For the main experiments, we use the default parameters for DPOK of reward weight $\beta = 10$ and KL weight = 0.01. For RTB we fix $\beta = 1.0$ for all prompts. We next perform an ablation over different values of $\beta$.

We plot in Table H.1 the final average reward and diversity score for models trained with different values of reward weight $\beta$ for the prompt "A green colored rabbit.". As expected, we find that increasing $\beta$ increases reward at the cost of diversity for RTB and DPOK. The exception is $\beta = 10$ for RTB which has slightly lower final reward than $\beta = 1$, which we could attribute to more difficult optimization due to the peaky distribution associated with higher reward weight.

Table H.1: Ablation of reward weights $\beta$ for "A green colored rabbit.".

| Model $\downarrow$ | $\beta \downarrow$ Metric $\rightarrow$ | Reward ($\uparrow$) | diversity ($\uparrow$) |
|---|---|---|---|
| Prior | - | -0.113 | 0.597 |
| DPOK (KL weight=0.01) | $\beta = 0.01$ | -0.27 | 0.1488 |
| | $\beta = 0.1$ | -0.06 | 0.1486 |
| | $\beta = 1.0$ | 0.638 | 0.1362 |
| | $\beta = 10.0$ | 1.492 | 0.076 |
| DDPO (KL weight=0.0) | $\beta = 10.0$ | 1.795 | 0.0493 |
| RTB | $\beta = 0.01$ | 0.485 | 0.1431 |
| | $\beta = 0.1$ | 1.525 | 0.0721 |
| | $\beta = 1.0$ | 1.756 | 0.0436 |
| | $\beta = 10.0$ | 1.568 | 0.0689 |

### H.1  Memory-efficient learning

We propose two methods to reduce the memory requirement of RTB fine-tuning.

**Stochastic subsampling.**   The expected gradient of the RTB objective (9) is unaffected by propagating gradient to a randomly sampled subset of the timesteps in a trajectory and rescaling by the inverse proportion of timesteps sampled. Stochastically subsampling timesteps for gradient propagation in this way can significantly decrease memory consumption because computation graphs for the remaining timesteps do not need to be maintained; however, such subsampling increases gradient variance, so it is preferable to keep gradients for as many timesteps as possible to fit in memory. For our text-to-image experiments, we found sampling 8 timesteps out of 50 to keep gradients was sufficient.

**Batched gradient computation.**   An important property of the RTB objective is that computing its gradient does not require storing the computation graph of all timesteps. The gradient of the RTB objective for a single trajectory is just the sum of per-step log-likelihood gradients scaled by the RTB residual:

$$\nabla_\phi \mathcal{L}_{\text{RTB}}(\tau; \phi) = 2 \left( \log \frac{Z_\phi}{r(\mathbf{x}_1)} + \sum_{i=1}^{T} \log \frac{p_\phi^{post}(\mathbf{x}_{i\Delta t} \mid \mathbf{x}_{(i-1)\Delta t})}{p_\theta(\mathbf{x}_{i\Delta t} \mid \mathbf{x}_{(i-1)\Delta t})} \right) \cdot \nabla_\phi \sum_{i=1}^{T} \log p_\phi^{post}(\mathbf{x}_{i\Delta t} \mid \mathbf{x}_{(i-1)\Delta t}).$$

Because the likelihood gradients can be accumulated during the forward pass, this allows for a batched gradient accumulation version of the update. For trajectory length (number of diffusion steps) $T$ and accumulation batch size (number of time steps receiving a gradient signal in each backward pass) $B$, the number of batched forward passes required scales as $\frac{T}{B}$.

---

[8]https://github.com/google-research/google-research/tree/master/dpok

Only the accumulation batch size $B$, not the trajectory length $T$, is constrained by the memory budget. This means we can easily scale training with large number of diffusion steps without increasing the variance of the gradient through stochastic subsampling, with training time growing linearly with number of time steps under a fixed memory budget. Although this method is not used in the main experiments presented here, preliminary experiments confirm these observations.

We highlight that both methods are not applicable to diffusion samplers based on differentiable simulation (*e.g.*, PIS and DDS), which need to store the entire computation graph of SDE integration. For these methods, the memory requirement scales linearly with the trajectory length.

## H.2 Generated images

### H.2.1 A green-colored rabbit

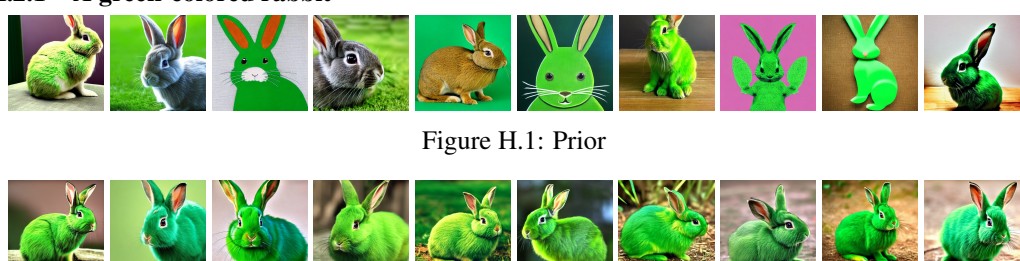

Figure H.1: Prior

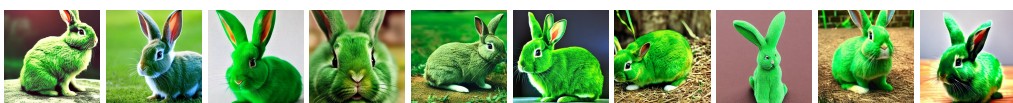

Figure H.2: DDPO

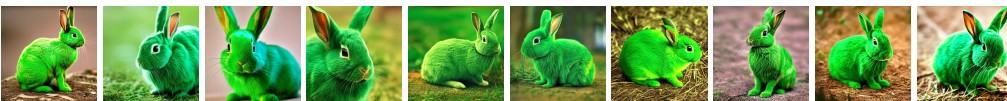

Figure H.3: DPOK

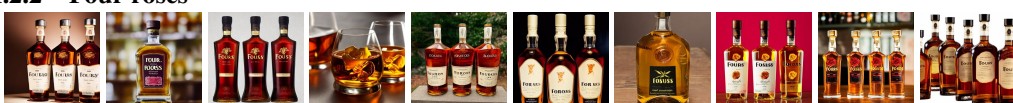

Figure H.4: RTB

### H.2.2 Four roses

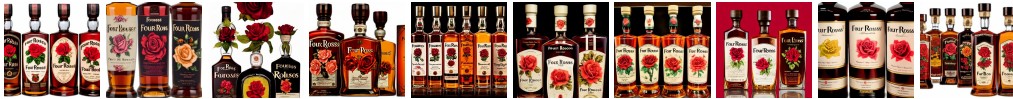

Figure H.5: Prior

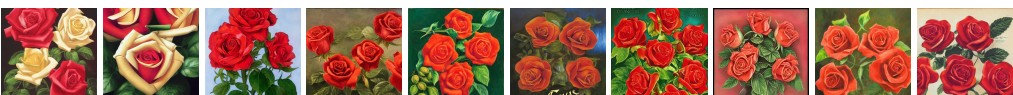

Figure H.6: DDPO

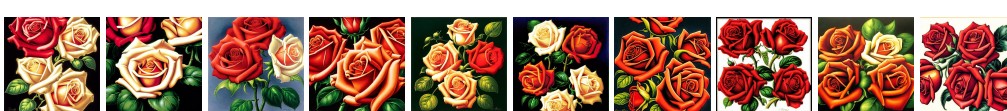

Figure H.7: DPOK

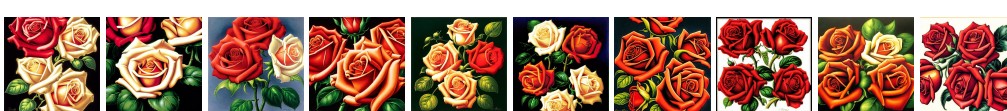

Figure H.8: RTB

### H.2.3 A cat and a dog

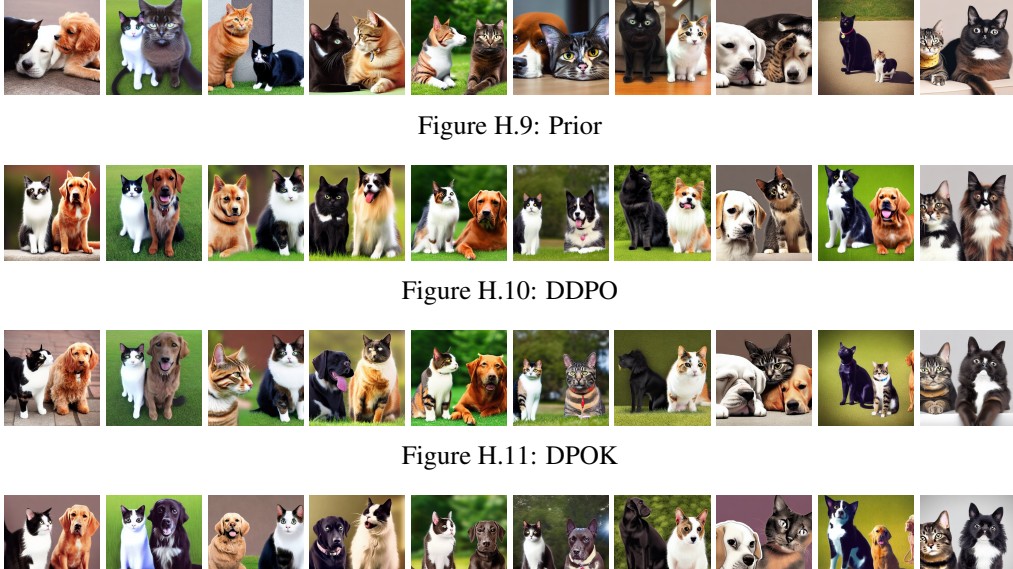

Figure H.9: Prior

Figure H.10: DDPO

Figure H.11: DPOK

Figure H.12: RTB

### H.2.4 A half - masked rugged laboratory engineer man with cybernetic enhancements as seen from a distance, scifi character portrait by greg rutkowski, esuthio, craig mullins.

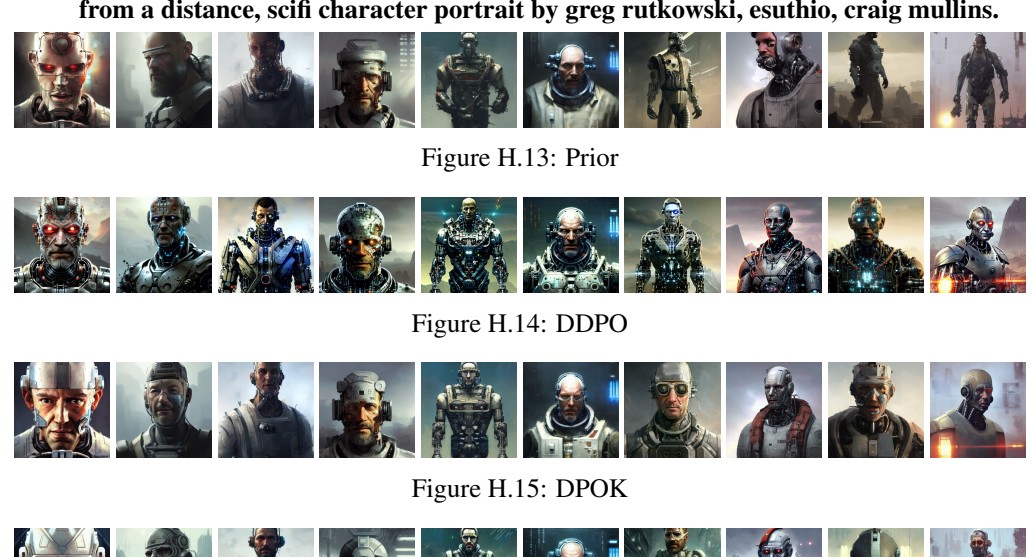

Figure H.13: Prior

Figure H.14: DDPO

Figure H.15: DPOK

Figure H.16: RTB

## I Compute resources

For classifier guidance experiments §3.1 we use train on a single NVIDIA V100 GPU. For text-conditional image generation §3.2, text infilling §F and offline §G, we use a single NVIDIA A100 large GPU. The total estimated compute time for all our experiments is 3000 hours.

