# OpenReview forum: "Amortizing intractable inference in diffusion models for vision, language, and control"
_NeurIPS.cc/2024/Conference — NeurIPS 2024 poster_

### Official Review · Reviewer_Z1Kk · 2024-07-11

**Soundness:** 3
**Presentation:** 3
**Contribution:** 3
**Rating:** 7
**Confidence:** 3

**Summary:**

This paper studies the problem of training diffusion models to sample from an intractable posterior distribution, defined by a prior diffusion model and an arbitrary likelihood function. The contributions are summarized as follows:

- This paper proposes relative trajectory balance (RTB) for training diffusion-based models to sample from a posterior distribution. RTB is derived from the perspective of continuous GFlowNets, which thus enables off-policy training. In contrast to related literature, the proposed approach performs posterior inference by fine-tuning a prior diffusion model in a parameter-efficient way.

- The effectiveness of the proposed approach is validated through experiments in vision, language modeling, and continuous control benchmarks.

**Strengths:**

- This paper proposes to train diffusion-based models to sample from an intractable posterior distribution. In particular, the proposed approach performs posterior inference by fine-tuning a prior diffusion model $p_{\theta}$ in a parameter-efficient way.

- The paper is well-written and well-organized.

**Weaknesses:**

We use a posterior sampler, parameterized by a prior diffusion model with a learnable drift term $u^{post}$, to simulate trajectories and obtain samples $x_{1}$. Then, we compute the likelihood function $r(x_{1})$ and the prior $p_{\theta}$. Using these as an unnormalized density, the model parameters can be updated using GFlowNet-based loss. For fine-tuning a diffusion model, do we need to ensure that the initial state is an empty set (discrete) or a point mass (continuous)? In general, the initial state for DDPM follows a Gaussian distribution. Does this violate the GFlowNet assumption?

**Questions:**

Some typos:

- Line 158: $x_{1} \leftarrow x_{0}$ --> $x_{1} \rightarrow x_{0}$?

- Eq11: $NN_{2}(x_{t}, t)$ --> $NN_{2}(t)$?

**Limitations:**

Yes. Limitations were included.

---

> ### Author Rebuttal · Authors · 2024-08-07
>
> Thank you for your feedback and positive assessment of the paper! We answer your question below:
>
> ### Does Gaussian initial state violate assumptions?
>
> Assuming the initial distribution is Gaussian does not violate the GFlowNet assumption, as it amounts to assuming the first generation step transitions from an abstract initial state to a sample from that Gaussian distribution. When training, e.g., with the trajectory balance objective, the likelihood of this first transition -- from the abstract initial state to the Gaussian noise sample -- appears in the decomposition of the sampling trajectory's forward likelihood. With RTB, the likelihood of this initial-to-noise step is the same for the prior and posterior models and thus cancels in the loss.
>
> ### Typos
>
> - The direction of the arrows in $x_1\leftarrow\dots\leftarrow x_0$ is intentional: the arrows always show the transitions that exist in the MDP, but the sampling of the noising trajectory happens in reverse (starting from data $x_1$ and ending at noise $x_0$). We can revise if you think this causes confusion.
> - Equation (11): Yes, ${\rm NN}_2$ isn't taking $x_t$ as input. We will fix this.
>
> Thanks for reading carefully!
>
> **Please let us know if there are any other questions we can answer during the discussion period.**

---

> > ### Comment · Reviewer_Z1Kk · 2024-08-12
> >
> > Thank you for your reply, and I will maintain my score.

---

### Official Review · Reviewer_VfXT · 2024-07-24

**Soundness:** 2
**Presentation:** 3
**Contribution:** 3
**Rating:** 5
**Confidence:** 2

**Summary:**

The paper looks at the problem of finetuning/training a generative model to sample from a desired posterior distribution when given access to a diffusion model prior. The experiments validated the generation capability across different tasks that diffusion models (and conditional generation) can be applied to - text infilling, text-to-image generation and training policies through offline RL. The work proposes a fine-tuning objective to train the model that will sample from the posterior (which in this case is instantiated from the prior’s checkpoint), which can be computed using off policy data/trajectories, ie. trajectories that are not all sampled from the posterior model.

**Strengths:**

Conditional generation is an active field of study and being able to use a foundational diffusion/generative model as a prior that can be employed/modified to get a posterior distribution of interest, is of interest in many applications.
The paper is mostly written clearly, and presents experiments on three different generation problems. Developing an off-policy objective for RL training/finetuning of a generative prior is very useful, however it’s somewhat unclear to me how off-policy the data actually is - I believe the data used is mostly on-policy with small amounts of noise added - could the authors add a small discussion making it very clear how the training data is related to the on/off-policy distributions of the prior/posterior models and how it is different from the data used to train the baselines, as well as how realistic these data collection assumptions are?

**Weaknesses:**

1. Speaking very qualitatively, the provided generations in Fig 4 and H.1 don’t look very different from each other and DPOK (and often even DDPO) seems to be generating samples of the same quality. This leads me to think that marginal improvements in the metrics provided (log likelihood etc) are not strongly tied to the generation quality. Also how were these samples chosen? To avoid cherry-picking and encourage reproducibility, i’d suggest mentioning the seed they were sampled with in the corresponding code readme files.
2. Why isn’t there a std dev/error reported for baseline numbers in Table 4? The numbers look close enough for many methods and aggregating metrics over seeds can significantly change results in RL.
3. A major concern is that the authors should ablate each stabilization/implementation trick they used when it is applicable to other baselines that they compare to (Clipping, using the pretrained ckpt from the prior for the finetuning, using parameter efficient training, any tweaks made to noising schedules, any special tricks used in data collection  etc) since there are so many that have been used.

Overall, I'm only moderately familiar with this area, and therefore will assign a lower confidence to my review. However, my current belief is that the paper has not demonstrated the advantages of the proposed method satisfactorily(or maybe this just has not been presented clearly) so far (either in terms of showing that it can really work well with very off-policy data that is easily available, or showing impressive gains in generation quality/diversity).

**Questions:**

1. On the text infilling task, am I correct in assuming that none of the other baselines had access to both x and y (the first 3 and the fifth sentences) since they are autoregressive and can’t condition on y? If yes, is the comparison to diffusion models that can condition on y fair? Should the comparison instead be with encoder-decoder or non-causal language models?
2. Is there a typo in Line 112? It says “... would sample the product distribution 𝑝post(x1) ∝ 𝑝𝜃 (x1)𝑟(x1). If 𝑟(x1) = 𝑝(y | x1) is a conditional distribution over another variable y, then 𝑝post is the Bayesian posterior 𝑝𝜃 (x1 | y).” Shouldn’t the product distribution be equal to the probably of the intersection of x and y and not y given x? Or is there a term which has implicitly been assumed to be constant/irrelevant?

**Limitations:**

The paper includes a very brief discussion on limitations which could be improved. I can provide more suggestions for this discussion after engaging with the authors and understanding their assumptions better.

---

> ### Author Rebuttal · Authors · 2024-08-07
>
> Thank you for your comments. We are happy to see that you appreciated the breadth of applications presented in our paper and the value of the contribution.
>
> Below we have attempted to answer your questions and to clarify a few misunderstandings in the review. We note that most of the listed weaknesses and questions pertain to specifics of individual experiments, not to the main contribution of the paper: a general-purpose method that can be applied to a wide range of tasks *simultaneously*, achieving results comparable to specialized methods that had been developed for each problem. We hope that our reponses will help to answer your questions and concerns.
>
> ### Improvements in image quality
>
> Measuring posterior sampling quality is difficult for text-to-image generation models, since we do not have ground truth samples from the posterior. We acknowledge that differences in visual quality compared to DPOK are not apparent, and the visual results (Figure 4) should be seen as complementing Figure 3 in showing comparable quality and diversity to baselines.
>
> RTB training facilitates asymptotically unbiased posterior sampling. This may not necessarily translate to improvements in auxiliary fidelity metrics such as those we must resort to when ground truth posterior samples are unavailable (as in the text-to-image experiments). However, in the experiments where we do have access to unbiased samples from the target classes (MNIST/CIFAR-10), it does translate to better FID scores (Table 2).
>
> In practice, text-to-image generation systems may use a combination of techniques to improve samples (e.g., collecting data from human raters, imposing auxiliary aesthetic constraints, etc.). Our work adds a novel, general, and probabilistically principled technique to this toolbox.
>
>
> ### Fair comparisons
>
> We answer a few questions related to comparisons between other methods and hours, hoping to clarify why we believe they are fair and informative.
>
> #### Infilling baselines seeing the right context
>
> Good question; no, it is **not** the case that baselines see only $x$ (left context) and not $y$ (right context). The fact that this was not clear suggests we should write it explicitly in the paper; thank you for directing our attention to it.
>
> In fact, all the baselines with the exception of "Prompt (**x**)" do condition on both $x$ and $y$, making the methods comparable. For example, for the prompting and SFT autoregressive baselines, the model receives a prompt of the form "Beginning: {x}, End: {y}, Middle: ", following prior work (see Appendix F). The difference between "Prompt (**x**)" and "Prompt (**x**,**y**)" shows the importance of conditioning on $y$.
>
> #### Algorithm components
>
> To clarify, we apply the same collection of techniques with RTB and with methods from prior work wherever applicable. For example, all methods used the same noising schedules. Gradient clipping, fine-tuning the pretrained prior checkpoint, and parameter-efficient training were done for RL baselines as well.
>
> On the other hand, some techniques used with RTB training are *not applicable* to other methods. Data collection methods used for off-policy training cannot be applied to compared to manifestly on-policy finetuning algorithms such as DPOK. The inference-time classifier guidance baselines, such as those in Table 2, do not require any training, so training hyperparameter choices do not apply to them.
>
> Altogether, while we believe the comparisons we make are fair, we agree that ablations on the techniques used in the paper (while not being applicable to baseline methods) would add to the understanding of the proposed algorithm. We will include ablation studies about this in the final version.
>
> #### Prior/posterior training data
>
> It is important to note that the newly proposed algorithms **and** most baselines are "data-free" -- that is, the prior model is pretrained and we receive no ground truth samples from the posterior, but rather must discover its modes through exploration and can query for the reward of a generated sample.
>
> #### How were samples for figures chosen?
>
> In the image generation figures (Figures 3 and 4 and Appendix H.1), the samples are not cherry-picked. In Appendix H.1, in each row the images are generated from seeds 0-9.
>
> #### RL baselines
>
> The reporting of baseline numbers and highlighting in Table 4 exactly matches the form in a line of previous offline RL papers (see, e.g., [35,43,36]), where std is reported only for the newly proposed algorithm alongside baseline means. However, we will update the paper to include std for comparisons, and have attached an updated table with std bars from baselines which report this metric in the global rebuttal document (Table 4).
>
> ### Line 112
>
> In that line, $p^{\rm post}$ is a distribution over $x_1$, where $y$ is fixed. For a fixed $y$, the product is proportional *as a function of $x_1$* to the posterior $p(x_1\mid y)$.
>
> **Thank you again for your feedback. We hope these answers and clarifications help; if they do, please consider updating your score. Let us know if there are any other questions we can answer during the discussion phase.**

---

> > ### Comment · Reviewer_VfXT · 2024-08-10
> >
> > Dear Authors,
> >
> > Thank you for your response.
> >
> > I had the following clarification questions/thoughts:
> > 1. " a general-purpose method that can be applied to a wide range of tasks simultaneously, achieving results comparable to specialized methods that had been developed for each problem." - is this true for all baselines considered in the work (that they are developed for specific problems)? Could the DPOK/DDPO baseline not have have been used on all the tasks? Could you explain why it was not applicable to the other two setups (Offline RL, text infilling)
> > 2. On offline RL the method is slightly worse than both QGPO and D-QL (if we count the number of setups where a method is not within the top 5%) - and so I would suggest removing the claim from the abstract that it achieves state-of-the-art results - instead saying that it almost matches the state-of-the-art (if the other two methods are that).

---

> ### Author Response · Authors · 2024-08-11
>
> Dear reviewer,
>
> Thanks for your response. We are happy to clarify these points:
> 1. The training free baselines (DPS, LGD-MC) cannot be used with discrete diffusion models for the text infilling task and are only derived for the particular case of inverse problems in continuous space. DPOK/DDPO are RL finetuning baselines which can in principle be used for all the tasks, but do require significant changes to be adopted for the case of discrete diffusion (which was not considered in the original work). Note that **all the baselines are also biased samplers**, and hence expected to give worse performance for tasks requiring unbiased posterior inference such as classifier guidance (as we show in the paper) and other important scientific tasks (for example [1]). In the context of offline RL, the D-QL baseline trains the policy with an objective very similar to DPOK - The policy is trained to maximize the Q function with a per-step KL regularization term with behavior policy. We can add some notes in the appendix to make these connections clearer in the final version.
> 2. We agree that the claim "matches state-of-the-art" is more accurate, and will update the abstract in the revision.
>
> **We hope we've addressed your questions and concerns. Feel free to reach out with any additional questions.**
>
> [1] Adam, Alexandre, et al. "Posterior samples of source galaxies in strong gravitational lenses with score-based priors." arXiv preprint arXiv:2211.03812 (2022).

---

> > ### Comment · Reviewer_VfXT · 2024-08-13
> >
> > Dear Authors,
> >
> > Thank you for your response.
> >
> > From what I understand, empirically, the results of RTB look similar to those of DPOK (or DPOK-like methods, ie, D-QL, like the authors said) on 2 of the 3 tasks: offline RL and text-to-image generation, and it was not tried out on the text-infilling task since that would require changes beyond what the DPOK paper had. Additionally, the qualitative examples/infills are not shown in the appendix for for the GFN baseline in the autoregressive prior category of models which seems to get the next best scores in table 3 so it is hard to say if the generated infills are much better than the next best performing method or not.
> >
> > The experimental results do not demonstrate an advantage to using this approach, but the method is more principled compared to alternative methods (I have not validated the proofs in the appendix rigorously and will consult the other reviewers in the discussion phase to follow about this) and I will update my score to reflect this.

---

### Official Review · Reviewer_4zDq · 2024-07-30

**Soundness:** 3
**Presentation:** 4
**Contribution:** 3
**Rating:** 5
**Confidence:** 4

**Summary:**

This paper proposed a method to train a posterior $p^{post}(x)$ given a prior distribution $p(x)$ and some additional (possibly unnormalized) constraint function $r(x)$, when the prior is a pretrained diffusion model. Through the choice over $r(x)$, this setup can capture a wide range of tasks. The authors propose *relative trajectory balance* (RTB) constraint, which, if satisfied by the posterior under training, guarantees that $p^{post} \sim p(x)r(x)$. By training $p^{post}(x)$ to satisfy the RTB constraint, one can recover the desired posterior. Importantly, this training objective allows for efficient computation of the loss and off-policy training. The paper provides convincing empirical results on a variety of tasks ranging from conditional generation to offline RL.

**Strengths:**

**Motivation and Theoretical Justification**
* The paper provides a theoretical justification (Proposition 1) for the proposed method, which is a natural extension of the trajectory balance constraint for the setting where we cannot easily sample from $p^{post}$. This is a realistic constraint that one often encounters when trying to perform posterior inference given a pretrained prior.

**Empirical Analysis**
* While it's clear that the formulation does capture a wide range of tasks, it is nice to see an experimental evidence of that. The generality of the proposed method is emphasized through the results on four different tasks.
* I appreciate that the authors took time to discuss some of the implementation details (Sec 2.3).

**Writing Quality**
* The presentation is very well-done. The problem being tackled is clear, and the technical aspects of the solution are intuitive and easy to follow.

**Weaknesses:**

**Scalability**
* One main concern I have is how scalable the method is due to trajectory matching, especially as the prior becomes larger.  The main benefit of this technique relies on the assumption that one aims to repurpose an existing diffusion prior, which is presumable a large foundation model -- so it's important that the method scales desirably to the prior's size, which is not clear from the results in the paper.
  * Section H discusses some techniques employed to reduce memory usage, but even this relatively small-scale experiment (50 steps on latent diffusion with LoRA), the authors report that they could fit only 8 timesteps on an A100.
  * It'd be very useful to get some idea of how sensitive the method is to these hyperparameters. One extreme case to consider would be to use single-step gradient subsampling for a much larger T (e.g. 1000 steps).

**Experimental Results**
* Sec 2.3 claims that the proposed method should easily generalize to conditional constraints (which I believe), but there doesn't seem to be any experiment that tests this claim. For example, it'd strengthen the paper to have a result where RTB training successfully extracts a class-conditional model (that takes a conditioning input) from an unconditional one (perhaps for the CIFAR experiment).
* The text-conditional image synthesis experiment seems to be done for only four hardcoded text prompts, which is pretty limited.
* While the text in-filling results are great to have, a more practical and interesting task would be fine tuning a pretrained LLM. As hinted in Sec. 2.4, I'm curious if RTB can be used to efficiently customize an LLM, where we deviate from the Gaussian transition kernel.
* Having a data-dependent reference as an upper bound on the performance of the proposed method would be useful (e.g. classifier guidance trained classifier trained on noisy data, or directly fine-tuning the prior itself on a specific task). This can tell us more about how much of the potential performance the model captures.

**Questions:**

* For the MNIST even-digit experiment, why was the reward function chosen to be $max_{i \textrm{ even}} p(c=i \mid x)$? Wouldn't $\sum_{i \textrm{ even}} p(c=i \mid x)$ be a more reasonable reward? I'm curious if this particular choice of reward makes it more likely for certain methods to mode collapse (e.g. by maximizing the probability for a single even class), and whether the results in Table 2 would change for MNIST with a different $r(x)$.
* Could you share some more details of how LoRA was used for MNIST/CIFAR setup (e.g. which weights were LoRAed)?  Also I'm curious what happens if you don't use LoRA and train the posterior on full set of weights (from prior weight or from scratch), as I LoRA could act as a regularizer.
* How long is the ratio between prior vs. posterior training in terms of flops or wall-clock time?  How quickly the posterior can be fine-tuned could limit the applicability of this method on certain real-world tasks.

**Limitations:**

Yes. For concerns and potential limitations, see the comments and questions above.

---

> ### Author Rebuttal · Authors · 2024-08-07
>
> Thank you for your constructive feedback and good questions. We are glad you found the paper well-written and the empirical results compelling. We believe that the diversity of domains on which we showed the effectiveness of our approach makes this work both a valuable illustration of off-policy finetuning for intractable diffusion posteriors and a starting point for a number of possible domain-specific applications.
>
> We have addressed each of your questions below and added three new experiments in response to your comments (please see the response to all reviewers).
>
> ### Text infilling and LLM fine-tuning
>
> We believe there may be a misunderstanding present. The infilling results are for the setting of *fine-tuning a pretrained discrete diffusion language model*, which does not use a Gaussian transition kernel. In discrete diffusion language models, the kernel is categorical at each position.
>
> The text infilling experiment (Table 3 in the paper) thus supports two claims:
> - that the proposed diffusion fine-tuning algorithm is applicable outside the Gaussian setting (and even outside the continuous setting);
> - that fine-tuning of LLMs for intractable posteriors, such as infilling, using off-policy RL can be generalized beyond autoregressive language models (as done in [28]).
>
> ### On scalability and cost
>
> We answer a few questions related to scaling and fine-tuning cost comparisons.
>
> #### Scalability
>
> RTB actually scales surprisingly well to larger models and more diffusion steps (relative to simulation-based methods) despite being a trajectory-level objective.
>
> An important observation about the RTB objective -- not included in our original submission -- is that computing the RTB gradient does not require storing the computation graph of all timesteps to be updated. We observe that the gradient of the RTB objective for a single trajectory is just the sum of per-step log-likelihood gradients scaled by the RTB residual: $$
> \nabla_{\phi}L_{RTB} = 2\left(\log\frac{Z_{\phi}}{r(x_1)} + \sum_{i=1}^T\log\frac{p_{\phi}^{post}(x_{i\Delta t} \mid x_{(i-1)\Delta t})}{p_{\theta}(x_{i\Delta t} \mid x_{(i-1)\Delta t})}\right) \cdot \nabla_{\phi}\sum_{i=1}^T\log p_{\phi}^{post}(x_{i\Delta t} \mid x_{(i-1)\Delta t}).$$ Because the likelihood gradients can be accumulated during the forward pass, this allows for a batched gradient accumulation version of the update. For trajectory length (number of diffusion steps) $T$ and accumulation batch size (number of time steps receiving a gradient signal in each backward pass) $B$, the number of batched forward passes required scales as $T/B$.
>
> Only the accumulation batch size $B$, not the trajectory length $T$, is constrained by the memory budget. This means we can easily scale training with large number of diffusion steps $T$ without increasing the variance of the gradient through stochastic subsampling, with training time growing only linearly with number of time steps under a fixed memory budget.
>
> The batched update implementation was not included in our original codebase, but we have performed experiments to validate it and will include them in the revision.
>
> Finally, we highlight that this is in contrast to other diffusion samplers (e.g., PIS/DDS/DIS) which differentiate through the sampling (SDE integration) process and therefore need to store the entire computation graph. For these methods, memory necessarily scales with the length of the trajectory.
>
> #### Cost of training posterior vs. prior
>
> For our image experiments (Table 2 and Figure 2), a very small number of finetuning steps is necessary for convergence to finetune a posterior with RTB (usually between 100 and 300 are sufficient, although we run for 1500), where each step performs a gradient update on a batch of trajectories (from 16 to 32, depending on the experiment). In contrast, training a prior diffusion model even on MNIST requires tens of thousands of training iterations.
>
> #### LoRA in MNIST/CIFAR setup
>
> The experiments on discrete diffusion and offline RL (Sections 3.3 and 3.4) do not use LoRA, while the image experiments (Sections 3.1 and 3.2) do. Note that parameter-efficient fine-tuning is standard for large models such as Stable Diffusion and also adopted by baseline methods (e.g., DPOK).
>
> For example, for MNIST and CIFAR we use LoRA (rank=32) on the key, query and value vectors of the denoiser’s attention mechanism of a UNet model. RTB can be applied without LoRA parameterization on these tasks too; please see the reponse to all reviewers and Table 1 in the attached pdf.
>
> ### Conditional constraints
>
> Using conditional constraints requires a more detailed study of architectures and conditioning strategies, since the unconditional prior cannot directly be finetuned. However, we have run a preliminary experiment on the MNIST even/odd digits task (Table 2); please see the response to all reviewers.
>
> ### Data-dependent upper bound
>
> Please see the response to all reviewers for such an upper bound on the text infilling task, where we have a larger ground truth dataset.
>
> ### Max or sum in classifier reward
>
> Thank you for the qustion. The choice of the reward function for the MNIST finetuning task (L220) was to discourage generation of ambiguous digits: for example,  $\sum_{i\text{ even}} p(c=i\mid x)$ is high when $p(c=0\mid x)\approx p(c=2\mid x)\approx\dots\approx p(c=8\mid x)$. In the current formulation, high reward can only be achieved by generating an image whose likelihood of at least one class is high. We observed that replacing max by sum resulted in more such ambiguous digits, which would appear to be a weakness of the pretrained prior model.
>
> ### Prompts for ImageReward task
>
> For the results in Figures 3 and 4, we use a similar setup to DPOK [15]. The prompts are copied verbatim from that paper.
>
> **We hope we have answered your questions satisfactorily; if so, please consider updating your score. Let us know if there are any other questions we can answer during the discussion phase.**

---

> > ### Author Response · Authors · 2024-08-12
> >
> > Dear Reviewer 4zDq,
> >
> > Thank you again for your initial feedback. We have addressed your concerns in the above comment and added three new experiments based on your suggestions in the global response.
> >
> > Would you kindly let us know if these have affected your evaluation of the paper and if there is any further clarification we can provide?
> >
> > The authors.

---

### Author Rebuttal · Authors · 2024-08-07

**We would like to thank all the reviewers for their comments. The reviewers all noted that the paper is well-written (4zDq, VfXT, Z1kk) and pointed out the the broad utility of the proposed approach (4zDq, VfXT, Z1kk), the theoretical justification for the proposed method (4zDq), its efficiency (Z1kk), and the quality of the empirical analysis (4zDq).**

The attached PDF file contains the following:
- **Table 1:** RTB on CIFAR without LoRA (suggestion by Reviewer 4zDq, ablation for Reviewer VfXT).
  - While LoRA was used in the oriignal experiments, full model finetuning can be achieved effectively, using a very small (~1e-5) learning rate to avoid instabilities. We have included results for CIFAR-10 in Table 1. LoRA, however, provides significant memory and speed advantages, including a reduced sensitivity to the learning rate (with LoRA we can train effectively with lr=5e-4 or even lr=1e-3). We will include all details and results in the final version.
- **Table 2:** Conditional constraints experiment for even/odd MNIST (suggestion by Reviewer 4zDq).
  - We use a naive parametrization of training the prior with an extra input channel, which we populate with 0 or 1 for even or odd conditioning during RTB posterior fine-tuning. We look forward to future work which develops more specialized architectures for handling conditional constraints.
- **Table 3:** Upper bound for text-infilling task (suggestion by Reviewer 4zDq).
  - We (supervised-)fine-tune the prior diffusion language model on the entire Stories dataset (which consists of 50k examples compared to the 1k pairs of left and right context used for training with RTB and other baselines). This serves as an upper bound on the performance of all the data-free methods, including RTB.
- **Table 4:** Standard deviation bars for offline RL baselines that report them in their papers (suggestion by Reviewer VfXT).

---

### Author Response · Authors · 2024-08-10
**Any more questions?**

Dear reviewers,

As we approach the end of the discussion period, we’d like to ask if we can give any final clarifications that would help you assess the paper.

Otherwise, if the responses and new results we shared earlier have resolved any doubts, please consider updating your ratings.

Thank you again for your time.

The authors

---

### Decision · Program_Chairs · 2024-09-25

**Decision:**

Accept (poster)

**Comment:**

This paper proposes Relative Trajectory Balance (RTB), a novel method for training diffusion models to sample from intractable posterior distributions by fine-tuning a pre-trained prior. Reviewers found the paper well-written (4zDq, VfXT, Z1kk) with a clear motivation and theoretical justification (4zDq). They appreciated the broad applicability of the method demonstrated through experiments across various domains (4zDq, VfXT, Z1kk).

Reviewers raised concerns regarding the scalability of RTB (4zDq) and the lack of certain experimental results, such as conditional constraints and data-dependent upper bounds (4zDq). They also questioned the significance of the empirical improvements, particularly in image generation quality (VfXT). The authors addressed these concerns in their rebuttal by providing additional experiments (conditional MNIST, text infilling upper bound) and clarifying the scalability of RTB through a batched gradient accumulation implementation. They also addressed the fairness of comparisons with baselines (VfXT) and provided further details on experimental setups (4zDq, VfXT). Although the question regarding the significance of empirical improvement still remains (VfXT).

While some reviewers initially expressed reservations about the empirical results (VfXT), the authors' clarifications and additional experiments paritally alleviated most of these concerns.  Reviewers ultimately found the method to be technically sound (4zDq, Z1kk) and offering a valuable contribution to the field (4zDq, VfXT, Z1kk). Leading to the decision of a weak acceptance.